# Evaluating China's fossil-fuel CO$_2$ emissions from a comprehensive dataset of nine inventories

Pengfei Han[1*], Ning Zeng[2*], Tom Oda[3], Xiaohui Lin[4], Monica Crippa[5], Dabo Guan[6,7], Greet Janssens-Maenhout[5], Xiaolin Ma[8], Zhu Liu[6,9], Yuli Shan[10], Shu Tao[11], Haikun Wang[8], Rong Wang[11,12], Lin Wu[4], Xiao Yun[11], Qiang Zhang[13], Fang Zhao[14], Bo Zheng[15]

[1]State Key Laboratory of Numerical Modeling for Atmospheric Sciences and Geophysical Fluid Dynamics, Institute of Atmospheric Physics, Chinese Academy of Sciences, Beijing, China

[2]Department of Atmospheric and Oceanic Science, and Earth System Science Interdisciplinary Center, University of Maryland, College Park, Maryland, USA

[3]Goddard Earth Sciences Research and Technology, Universities Space Research Association, Columbia, MD, United States

[4]State Key Laboratory of Atmospheric Boundary Layer Physics and Atmospheric Chemistry, Institute of Atmospheric Physics, Chinese Academy of Sciences, Beijing, China

[5]European Commission, Joint Research Centre (JRC), Directorate for Energy, Transport and Climate, Air and Climate Unit, Ispra (VA), Italy

[6]Department of Earth System Science, Tsinghua University, Beijing, China

[7]Water Security Research Centre, School of International Development, University of East Anglia, Norwich, UK

[8]State Key Laboratory of Pollution Control and Resource Reuse, School of the Environment, Nanjing University, Nanjing, China

[9]Tyndall Centre for Climate Change Research, School of International Development, University of East Anglia, Norwich, UK

[10]Energy and Sustainability Research Institute Groningen, University of Groningen, Groningen 9747 AG, Netherlands

[11]Laboratory for Earth Surface Processes, College of Urban and Environmental Sciences, Peking University, Beijing, China

[12]Department of Environmental Science and Engineering, Fudan University, Shanghai, China

[13]Ministry of Education Key Laboratory for Earth System Modeling, Department of Earth System Science, Tsinghua University, Beijing, China

[14]Key Laboratory of Geographic Information Science (Ministry of Education), School of Geographic Sciences, East China Normal University, Shanghai, China

[15]Laboratoire des Sciences du Climat et de l'Environnement, CEA-CNRS-UVSQ, UMR8212, Gif-sur-Yvette, France

*Correspondence to*: Pengfei Han (pfhan@mail.iap.ac.cn); Ning Zeng (zeng@umd.edu)

**Abstract.** China's fossil-fuel CO$_2$ ~~emissions~~ (FFCO$_2$) emissions account~~ed~~ for ~~about~~ approximately 28% of the global total FFCO$_2$ in 2016. An accurate estimate of China's FFCO$_2$ emissions is a prerequisite for global and regional carbon budget analyses and the monitoring of carbon emission reduction efforts. However, ~~large~~ significant uncertainties and discrepancies exist in estimations of China's FFCO$_2$ emissions ~~estimations~~ due to a lack of detailed traceable emission factors (EF) and multiple statistical data sources. Here, we evaluated China's FFCO$_2$ emissions from nine~~9~~ published global and regional emission datasets. These datasets show that the total emissions increased from 3.4 (3.0-3.7) in 2000 to 9.8 (9.2-10.4) Gt CO$_2$ yr$^{-1}$ in 2016. The variations in ~~their~~ these estimates were ~~due~~ largely due to the different EF (0.491-0.746 t C per t of coal) and activity data. The large-scale patterns of gridded emissions showed a reasonable agreement with high emissions being

concentrated in major city clusters, and the standard deviation mostly ranged from 10-40% at the provincial level. However, patterns beyond the provincial scale ~~vary~~ varied ~~greatly~~ significantly, with the top 5% of the grid-level accounting for 50-90% of total emissions ~~for~~ in these datasets. Our findings highlight the significance of using locally- measured EF for ~~the~~ Chinese coals. To reduce ~~the~~ uncertainty, we ~~call on the enhancement of~~ recommend using physical $CO_2$ measurements and use ~~them~~ these values for datasets validation, key input data sharing (e.g.~~,~~ point sources) and finer resolution validations at various levels.

**Keywords**: fossil-fuel $CO_2$ emissions, spatial disaggregation, emission factor, activity data, comprehensive dataset

## 1 Introduction

Anthropogenic emissions of carbon dioxide ($CO_2$) is one of the major ~~contributions in accelerating~~ accelerators of global warming (IPCC, 2007). ~~The g~~Global $CO_2$ emissions from fossil fuel combustion and industry processes increased to 36.23 Gt $CO_2$ $yr^{-1}$ in 2016, with a mean growth rate of 0.62 Gt $CO_2$ $yr^{-1}$ ~~per year~~ over the last decade (Le Quéré et al., 2018). In 2006, China became the world's largest emitter of $CO_2$ (Jones, 2007). ~~The~~ $CO_2$ emissions from fossil fuel combustion and cement production ~~of~~ in China ~~was~~ were 9.9 Gt $CO_2$ in 2016, accounting for ~~about~~ approximately 28% of all global fossil-fuel based $CO_2$ emissions (Le Quéré et al., 2018;IPCC AR5, 2013). To avoid the potential adverse effects from climate change (Zeng et al., 2008;Qin et al., 2016), the Chinese government has pledged to peak its $CO_2$ emissions by 2030 or earlier and to reduce ~~the~~ $CO_2$ emissions per unit gross domestic product (GDP) by 60-65%~~.~~ ~~below~~ less than the 2005 levels (SCIO, 2015). Thus, an accurate quantification of China's $CO_2$ emissions is the first step ~~in~~ toward understanding its carbon budget and making carbon control policy.

~~Chinese~~ The total emission estimates for China are thought to be uncertain or biased due to the lack of reliable statistical data and/or the use of generic emission factors (EF) (e.g.~~,~~ (Guan et al., 2012); (Liu et al., 2015b)). National and provincial data- based inventories use~~d~~ activity data from different sources. The Carbon Dioxide Information Analysis Center (CDIAC) use~~s~~d national energy statistics from the United Nations (UN) (Andres et al., 2012), and both the Open-Data Inventory for Anthropogenic Carbon D~~d~~ioxide (ODIAC) and Global Carbon Project (GCP) mainly use CDIAC total estimates~~.~~ and thus~~,~~ they are identical in time series (Le Quéré et al., 2018;Oda et al., 2018). The Emissions Database for Global Atmospheric Research (EDGAR) and Peking University $CO_2$ (PKU-$CO_2$, hereafter named ~~as~~ PKU) derive~~d~~ emissions from the energy balance statistics of the International Energy Agency (IEA) (Janssens-Maenhout et al., 2019a;Wang et al., 2013). ~~On the other hand~~In contrast, ~~the~~ provincial data- based inventories developed within China all use~~d~~ the provincial energy balance sheet ~~in~~ from the China Energy Statistics Yearbook (CESY)~~,~~ ~~from~~ National Bureau of Statistics of China (NBS) (Cai et al., 2018;Liu et al., 2015a;Liu et al., 2013;Shan et al., 2018). ~~As for EF, th~~There are generally four sources of EF, i.e., 1) The Intergovernmental Panel on Climate Change (IPCC) default values~~.~~ which ~~that has~~ have been adopted by ODIAC and EDGAR (Andres et al., 2012;Janssens-Maenhout et al., 2019b;Oda et al., 2018); 2) National Development and Reform

Commission (NDRC) (NDRC, 2012b); 3) China's National Communication, which reportsed to the United Nations

Framework Convention on Climate Change (UNFCCC) (NDRC, 2012a); and 4) The China Emission Accounts and Datasets

(CEADs) EF, which that are locally optimized through large sample measurements (Liu et al., 2015b). The existing estimates

of global total $FFCO_2$ emissions are comparable in magnitude, with an uncertainty that is generally within ±10% (Le Quéré

et al., 2018;Oda et al., 2018). However, there are great significant differences in these values at the national scale (Marland et

al., 2010;Olivier et al., 2014), with the uncertainty ranging from a few percent to more than 50% in the estimated emissions

for individual countries (Andres et al., 2012;Boden et al., 2016;Oda et al., 2018).

    Along with the total emissions estimates, the spatial distribution of emissions are is also important for several reasons: 1)

Spatial gridded products provide enhance our basic understandings on of $CO_2$ emissions; 2) They Sspatial distributions are

key inputs (as priors) for transport and data assimilation models, and which influenced the carbon budget (Bao et al., 2020);

and 3) For high- emissions areas recognized by multiple inventories, they spatial distributions can be used for policy making

in toward emissions reductions and can provide useful information for the deployment of instruments in emissions

monitoring for high-emissions areas recognized by multiple inventories (Han et al., 2020). At the global level, gridded

emissions datasets are often based on the disaggregation of country- scale emissions (Janssens-Maenhout et al., 2017;Wang

et al., 2013). Thus, the gridded emissions data are subjected to errors and uncertainties from due to the total emissions

calculations and emissions spatial disaggregation (Andres et al., 2016;Oda et al., 2018;Oda and Maksyutov, 2011). For

example, the Carbon Dioxide Information Analysis Center (CDIAC) distributes national energy statistics at a resolution of

1°×1° using the population density as a proxy (Andres et al., 2016;Andres et al., 2011). Further, to improve the spatial

resolution of the emissions inventory, the Open-Data Inventory for Anthropogenic Carbon dioxide (ODIAC) distributes

national emissions based on CDIAC and BP statistics with satellite nighttime lights and power plant emissions (Oda et al.,

2018;Oda and Maksyutov, 2011). (EDGAR ) derivesd emissions from the energy balance statistics of the International

Energy Agency (IEA), and obtains country- specific activity datasets from BP plc, United States Geological Survey (USGS),

World Steel Association, Global Gas Flaring Reduction Partnership (GGFR)/U.S. National Oceanic and Atmospheric

Administration (NOAA) and International Fertilizer Association (IFA). Gridded emissions maps at a resolution of 0.1°×x0.1°

were are produced using spatial proxy data based on the population density, traffic networks, nighttime lights and point

sources, as described in Janssens-Maenhout et al. (2017). Based on the sub subnational fuel data, population and other

geographically resolved data, a high-resolution inventory of global $CO_2$ emissions was developed at Peking University

(Wang et al., 2013). .

    In order toTo accurately calculate emissions, a series of efforts have been conducted to quantitatively evaluate China's $CO_2$

emissions using national or provincial activity data, local EF, and detailed data-sets of point sources (Cai et al., 2018;Li et al.,

2017;Wang et al., 2013). The China High Resolution Emission Database (CHRED) was developed by Cai et al. (2018) and

Wang et al. (2014) based on ~~the~~ provincial statistics, traffic network, point sources and industrial and fuel-specific EF. CHRED was featured ~~by~~ based on its exclusive point source data ~~for~~ from 1.58 million industrial enterprises from the First China Pollution Source Census. The ~~Mutli~~Multi-resolution Emission Inventory for China (MEIC) was developed by Zhang et al. (2007), Lei et al. (2011) and Liu et al. (2015a) at Tsinghua University through ~~integrating~~ the integration of provincial statistics, unit-based power plant emissions, population density, traffic networks~~,~~ and ~~emission factor (EF)~~EF (Li et al., 2017;Zheng et al., 2018b;Zheng et al., 2018a). The MEIC use~~d~~s the China Power Emissions Database (CPED), ~~and~~ in which the unit-based approach is used to calculate emissions for each coal-fired power plant in China with detailed unit-level information (e.g., coal use, geographical coordinates). ~~For the~~Regarding mobile emissions sources, a high-resolution mapping approach is adopted to constrain ~~the~~ vehicle emissions using a county-level activity database. CEADs was constructed by (Shan et al., 2018;Shan et al., 2016) and Guan et al. (2018) based on different levels of inventories to provide emissions at the national and provincial scales. CEADs use~~d~~s coal EF from ~~the~~ large-sample measurements (602 coal samples and samples from 4,243 coal mines)~~. And this is~~, which are assumed to be more accurate than the IPCC default EFs.

~~However, r~~Regardless of these efforts, ~~however, the amount of~~ China's $CO_2$ emissions remain~~s~~ uncertain due to the large discrepancy among current estimates, of which the difference ranges from 8-24% of ~~the~~ total estimates (Shan et al., 2018;Shan et al., 2016). Several studies have ~~made~~undertaken efforts ~~of quantifying~~to quantify the possible uncertainty in China's $FFCO_2$, such as differences ~~from~~ due to estimation approaches (Berezin et al., 2013), energy statistics (Hong et al., 2017;Han et al., 2020), spatial scales (Wang and Cai, 2017)~~,~~ and point source data~~-~~. Importantly, the authors ~~would like to point out~~note that the lack of a comprehensive understanding and comparison of the potential uncertainty in estimates of China's $FFCO_2$, including spatial, temporal, proxy~~,~~ and magnitude components, ~~makes~~ causes Chinese emissions ~~possible data to be~~ more uncertain, and thus~~,~~ it is important to present, analyze and explain such differences among inventories~~-~~.

Here~~,~~ we evaluated the uncertainty in China's $FFCO_2$ estimates by synthesizing global gridded emissions datasets (ODIAC, EDGAR~~,~~ and PKU) and China-specific emission maps (CHRED, MEIC~~,~~ and the Nanjing University $CO_2$ (NJU) emission inventory). Moreover, several other inventories were used in the evaluation analysis, such as the Global Carbon Budget from the Global Carbon Project~~,~~ and the National Communication on Climate Change of China (NCCC).

The ~~purposes~~aims of this study were to: 1) ~~Q~~quantify the magnitude and the uncertainty in China's $FFCO_2$ estimates using the spread of values from ~~the~~ state-of-the-art inventories; 2) identify the spatiotemporal differences of China's $FFCO_2$ emissions ~~between~~ among the existing emission inventories and explore the underlying reasons for such differences. To our knowledge, this is the first comprehensive evaluation of the most up-to-date and ~~mostly~~ predominantly publicly available carbon emission inventories for China.

## 2. Emissions data

~~The~~ An evaluation analysis was conducted from 9 inventories including six~~6~~ gridded datasets (listed in Table 1, ODIAC, EDGAR, PKU, CHRED, MEIC, and NJU) and three~~3~~ other datasets (GCP/CDIAC, CEADs, and NCCC) containing statistical data. We selected the year 2012 for spatial analysis ~~since~~ because this is the most recent year available for all the gridded data-sets and also because ~~this is~~2012 was a peak year of emissions due to the strong reductions ~~from~~ following the impact~~s~~ of the 12th-Five-Year-Plan. Specifically, the global fossil fuel $CO_2$ emission~~s~~ datasets included the year 2017 version of ODIAC (ODIAC2017), ~~the~~ version ~~v~~4.3.2 of EDGAR (EDGARv4.3.2) and~~,~~ PKU-$CO_2$, all of which ~~all~~ use~~d~~ the Carbon Monitoring for Action (CARMA) as the point source. The China-specific emission~~s~~ data used were ~~the dated from the year~~from 2007 ~~of~~ from CHRED, ~~the~~ MEIC v1.3 and~~,~~ NJU-$CO_2$ v2017, all of which ~~all~~ use~~d~~ China Energy Statistical Yearbook (CESY) activity data. Moreover, three~~3~~ inventories were used as ~~a~~ reference~~s~~, i.e., GCP/CDIAC, CEADs and NCCC, ~~since~~ because GCP and ODIAC use~~d~~ CDIAC for ~~most~~ the majority of the years, except ~~for~~ the most~~-~~recent two years, ~~that~~ which were extrapolated ~~by~~ using BP data~~,~~. ~~t~~These three inventories were treated as inventory ~~one~~ in a time series comparison. Data were collected from the official websites ~~for~~ of ODIAC, EDGAR, PKU, and 6 six tabular statistical data sets, and were also acquired from ~~their~~ the authors ~~for of~~who developed CHRED, MEIC and NJU. See the supporting information for more details ~~on~~ of the data sources and the methodology ~~of~~used for each dataset.

## 3. Methodology for **the** evaluation of multiple datasets

We evaluated ~~these~~ the abovementioned datasets from three aspects: data sources, boundary (emission sectors) and methodology (Figure 1, Table 1 and S1, S2). ~~For~~ In regard to the data source, there are two levels: national data, such as UN or IEA statistics, and provincial~~-~~-level data, such as CESY. The emission sectors mainly include fossil fuel production, industry production and processes, households, transportation, aviation/shipping, agriculture, natural biomass burning from wild fire~~s~~ and the waste ~~for~~ from these datasets, ~~and where~~; Table S1 list~~s~~ the~~ed~~ sectors included in each inventory. ~~And~~ In addition, for methodology, the analysis of the inventories includes the total estimates (activity data and EF) aspect and the spatial disaggregation of point, line and area sources. ~~As~~ Fig. 1 ~~depicts~~shows ~~depicted~~ the conceptual procedure followed ~~in~~ for the total emissions estimates and how the gridded maps ~~are~~ were produced for all the inventories, and thus, it is important to know the differences in the activity data, EF and spatial proxy data and spatial disaggregation methods ~~they~~ used by previous scholars, to understand the differences among the inventories in regard to total emissions estimates and spatial characteristics.

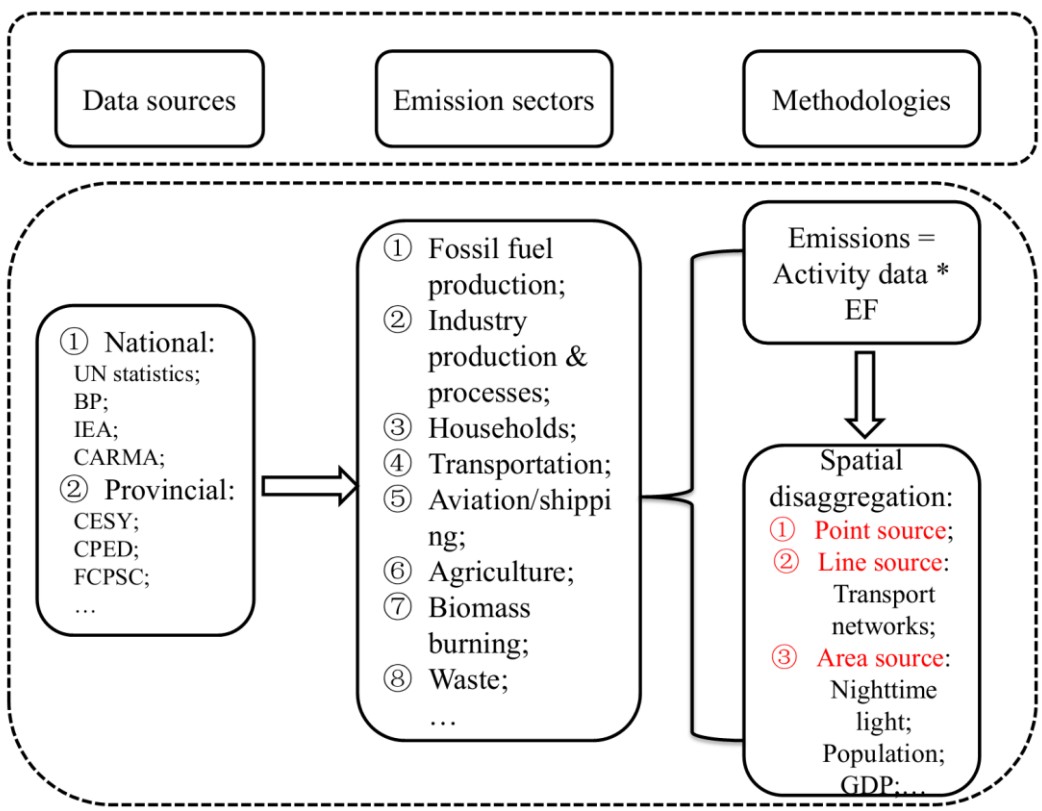

Figure 1 Conceptual diagram for data evaluation based on data sources, emission sectors and methodologies.

~~Preprocessing~~ The preprocessing of six gridded $CO_2$ emission datasets included several steps, which ~~that~~ are described as follows. First, ~~tThe~~ global map(s) of $CO_2$ emissions (i.e., ODIAC, EDGAR and PKU) were re-projected ~~to~~ using the Albers Conical Equal Area projection (that of CHRED). ~~And~~ Next, the nearest neighbor algorithm was used to resample different spatial resolution(s) into a pixel size of 10 km ~~x by~~ 10 km, and this method takes the value from the cell closest to the transformed cell as the new value. Second, the national total emissions were derived using the ArcGIS zonal statistics tool

~~for~~ from CHRED, while the other ~~others~~ emissions were from tabular data provided by the data owners. Finally, the grids for each inventory were sorted in ascending order and then plotted on a logarithmic scale to represent the distribution of emissions. To identify the contribution of high emission grids, emissions at the grid level that exceeded 50 kt $CO_2$ $yr^{-1}$ $km^{-2}$ and the top 5% emitting grids were selected for analysis.

Table 1 General information for of the emissions data-sets*

| Data | ODIAC2017 | EDGARv432 | PKU | CHRED | MEIC | NJU | CEADs | GCP/CDIAC | NCCC |
|---|---|---|---|---|---|---|---|---|---|
| Domain | Global | Global | Global | China | China | China | China | Global | China |
| Temporal coverage | 2000-2016 | 1970-2012 | 1960-2014 | 2007, 2012 | 2000-2016 | 2000-2015 | 1997-2015 | 1959-2018 | 2005, 2012, 2014 |
| Temporal resolution | Monthly | Annual | Monthly | Biennially or triennially | Monthly | Annual | Annual | Annual | Annual |
| Spatial resolution | 1 km | 0.1 degree | 0.1 degree | 10 km | 0.25 degree | 0.25 degree | N/A | N/A | N/A |
| Emission estimates | Global & National | Global & National | Global & National | National & Provincial | National & Provincial | National & Provincial | National & Provincial | Global & National | National |
| Emission factor for raw coal (tC per t of coal) | 0.746 | 0.713 | 0.518 | 0.518 | 0.491 | 0.518 | 0.499 | 0.746 | 0.491 |
| Uncertainty | 17.5% (95% CI) | ±15% | ±19% (95% CI) | ±8% | ±15% | 7-10% (90% CI) | -15% - 25% (95% CI) | 17.5% (95% CI) | 5.40% |
| Point | CARMA | CARMA3.0 | CARMA2.0 | FCPSC | CPED | CEC;A | N/A | N/A | N/A |

| | | | | | | | | | |
|---|---|---|---|---|---|---|---|---|---|
| source | 2.0 | | | | | CC;CCTEN | | | |
| Line source | N/A | the OpenStreetMap and OpenRailwayMap, Int. aviation and bunker | N/A | The national road, railway, navigation network, ~~and~~ traffic flows | Transport networks | N/A | N/A | N/A | N/A |
| Area source | Nighttime light | Population density, nighttime light | Vegetation and population density, nighttime light | Population density, land use, human activity | Population density, land use | Population density, GDP | N/A | N/A | N/A |
| Version name | ODIAC2017 | EDGARv4.3.2_FT2016,EDGARv4.3.2 | PKU-CO2-v2 | CHRED | MEIC v.1.3 | NJU-CO$_2$v2017 | CEADs | N/A | N/A |
| Year published/ updated | 2018 | 2017 | 2016 | 2017 | 2018 | 2017 | 2017 | 2019 | 2018 |

| | | | | Data developer | Data developer | Data developer | http://www.ceads.net/ (registration required) | https://www.globalcarbonproject.org/carbonbudget/19/data.htm | https://unfccc.int/sites/default/files/resource/China 2BUR_English.pdf |
|---|---|---|---|---|---|---|---|---|---|
| Data sources | http://db.cger.nies.go.jp/dataset/ODIAC/ | http://edgar.jrc.ec.europa.eu/overview.php?v=432_GHG&SECURE=123 | http://inventory.pku.edu.cn/download/download.html | Data developer | Data developer | Data developer | | | |
| References | Oda (2018) | Janssens-Maenhout (2017) | Wang et al., 2013 | Cai et al. (2018); Wang et al. (2014) | Zheng (2018); Liu et al. (2015) | Liu (2013) | Shan et al. (2018) | Friedlingstein et al. (2019) | NCCC (2018) |

* CI: Confidence interval; FCPSC: the First China Pollution Source Census; CPED: China Power Emissions Database; CEC: Commission for Environmental Cooperation;

ACC: China Cement Almanac; CCTEN: China Cement Industry Enterprise Indirectory; GDP: Gross domestic product; N/A: Not available.

# 4. Results

## 4.1 Total emissions and recent trends

The interannual variations of China's $CO_2$ emissions from 2000 to 2016 were evaluated from six~~6~~ gridded emission maps and three~~3~~ national total inventories (Figure 2). All the datasets show a significant increasing trend in the period of 2000 to 2013 from 3.4 to 9.9 Gt $CO_2$. The range of the nine~~9~~ estimates increased simultaneously from 0.7 to 2.1 Gt $CO_2$ (both are 21% of the corresponding years' total emissions). In the second period (from 2013 to 2016), the temporal variations mostly levelled off or even decreased. Specifically, the emissions estimated from PKU and CEADs showed a slight downward trend, even though ~~although~~ they used independent activity data ~~of~~ from IEA (2014) and National Bureau of Statistics (2016), and this downward trend ~~is~~ was attributed to changes in the industrial structure, improved combustion efficiency, emissions control and slowing economic growth (Guan et al., 2018;Zheng et al., 2018a).

There is a large discrepancy among the current estimates, ranging from 8.0 to 10.7 Gt $CO_2$ in 2012. NJU ~~has~~ had the highest emissions during the periods of 2005~~-~~—2015, followed by EDGAR, MEIC and CDIAC/GCP/ODIAC, while CEADs (National) and PKU were ~~much~~ significantly lower (Figure 2). ~~This~~ These discrepancies are ~~is~~ mainly because of three reasons: 1) the EF for raw coal was ~~higher~~ greater for EDGAR and ODIAC than the other databases~~s~~. The EFs were different for different fossil fuel types and cement production (Table S2). ~~Since~~ Because coal consumption ~~consisted~~ constituted 70-80% of total emissions, the coal EF ~~is~~ was more significant than the others. The EFs were different for the three major fossil fuel types (raw coal, oil and natural gas) and cement production (Table 1 and S2). ~~And~~ In addition, ~~they~~ the EFs ~~are~~ were obtained from either the IPCC default values or local optimized values from different sources. ~~They~~ The EFs do not change over time in these inventories, although they should, due to the unavailability of EFs over time; 2) differences in activity data, i.e., NJU, MEIC and CEADs (Provincial) use~~d~~ provincial data from CESY (2016), while CEADs (National) and~~,~~ PKU use~~d~~ national data from CESY (2016) and IEA (2014), respectively (Table 1 and S1), ~~and~~ such that the sum of provincial emissions ~~would be~~ is higher than the national total; and 3) differences in emission definitions (Table 1 and S1, emissions sectors). Although we tried to ~~make~~ ensure that these datasets ~~would be~~ as comparable as possible, ~~there are still~~ nonetheless minor differences in emission~~s~~ sources (sectors) remained. For example, EDGAR contains abundant industry process-related~~s~~ emissions, ~~while~~ whereas CEADs only consider~~s~~ed cement production (Janssens-Maenhout et al., 2019b). EDGAR and MEIC have~~a~~ similar trend~~s~~, ~~but~~ except for their magnitude~~s~~, ~~where~~ and MEIC is usually ~~higher~~ greater than EDGAR. This is a combined effect of the above three reasons. Moreover, MEIC use~~s~~ ~~the~~d provincial energy data from CESY (2016), ~~while~~ whereas EDGAR use~~s~~d ~~the~~ national--level data from IEA (2014). ~~But~~ However, MEIC's EF is lower than that of EDGAR. These opposing effects ~~would~~ bring ~~them~~ the data~~-~~sets closer in magnitude. ~~The~~ Both the gridded

products (ODIAC, EDGAR, MEIC and NJU) and national inventory (GCP/CDIAC) ~~both~~ show small differences in the magnitude of total emissions estimates and trend~~s~~ from 2000~~-~~—2007, ~~and~~ where the differences in magnitude increase~~sd~~ gradually from 2008 onward. Although the range increase~~sd~~ with time, the relative difference remains at ~~around~~ approximately 21% of the corresponding years' total estimates, indicating potential~~ly~~ systematic~~al~~ differences, such as the fact that EFs remain stable.

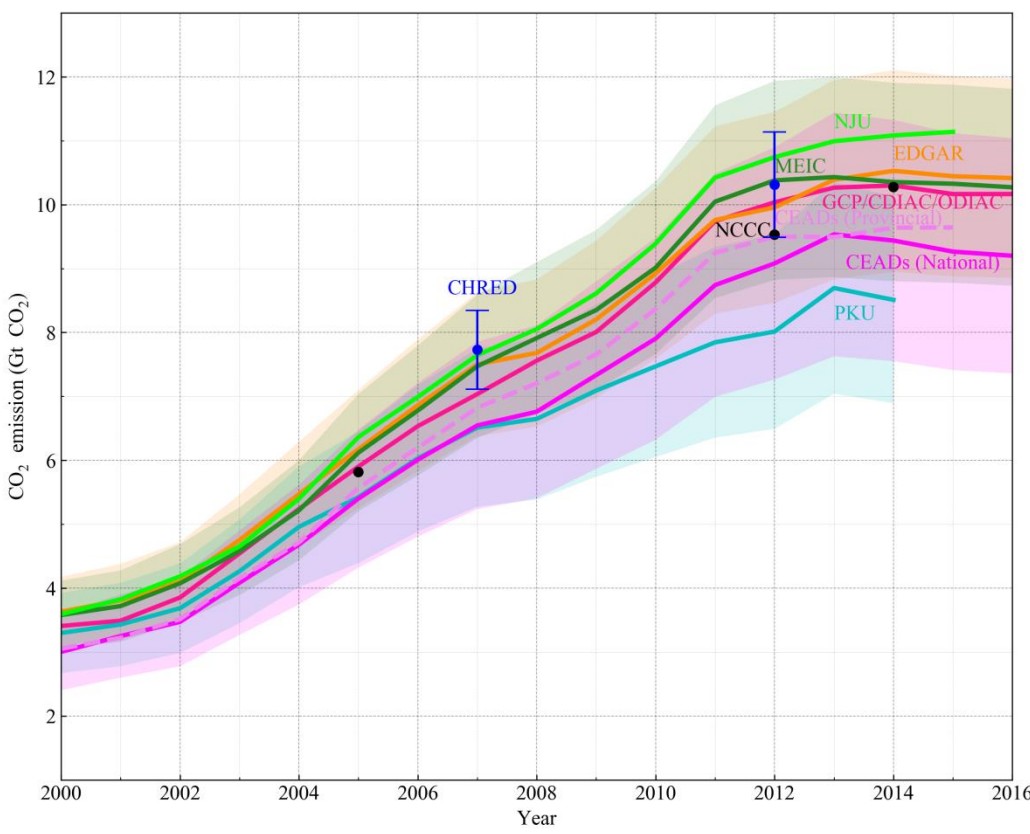

**Figure 2.** China's total FFCO$_2$ emissions from 2000 to 2016. The emissions are from the combustion of fossil fuels and cement production from different sources (EDGARv4.3.2_FT2016 includes international aviation and marine bunkers emissions). To ~~keep~~ maintain comparability and avoid differences result~~eing~~d from ~~the~~ emissions disaggregation~~-~~ (e.g.~~.~~ Oda et al. 2018(Oda et al., 2018)), the values ~~for~~ of the six~~6~~ gridded emission inventories are tabular data provided by the data developers before spatial disaggregation. Prior to 2014, GCP data ~~was~~ were taken from CDIAC, and those from 2015-2016 ~~was~~ were calculated based on BP data and the fraction of cement production emissions in 2014. ~~S~~The sha~~de~~ding area (error bar for CHRED) indicates uncertainties from the coauthors' previous studies (See Table 1).

## 4.2 Spatial distribution of FFCO$_2$ emissions

The evaluation of spatially~~-~~ explicit FFCO$_2$ emissions is fundamentally limited by the lack of direct physical measurements ~~on~~ at the grid scale~~s~~ (e.g.~~.~~ (Oda et al., 2018)). ~~We thus~~Thus, we attempted to characterize the spatial patterns of China's carbon emissions by presenting the available emissions estimate~~s~~ ~~available~~. We compared six~~6~~ gridded products, including ODIAC, EDGAR, PKU, CHRED, MEIC and NJU~~.~~ ~~in~~ for the year 2012~~. The~~, which ~~year 2012~~ was the most recent year for which all ~~the~~ six datasets were available. Spatially, the CO$_2$ emissions from the different datasets are concentrated in eastern China (Figure 3). ~~High~~The high-~~-~~emission areas were mostly distributed in city clusters (e.g.~~.~~ BeijingTianjin-Hebei

(Jing-Jin-Ji), the Yangtze River Delta, and the Pearl River Delta) and densely populated areas (e.g., the North China Plain, the Northeast China Plain and Sichuan Basin). These major spatial patterns are primarily due to the use of spatial proxy data, and are also in accordance with previous studies (Guan et al., 2018;Shan et al., 2018). However, there were notable differences among the different estimates at finer spatial scales. Large carbon emissions regions were found in the North China Plain and the Northeast China Plain for ODIAC (Figure 3a), PKU (Figure 3c), MEIC (Figure 3e) and NJU (Figure 3f), which ranged from 1000 to 10,000 t $CO_2$/km$^2$. However, high levels of emissions located in the Sichuan Basin were found from PKU, MEIC and NJU but not from ODIAC. This discrepancy in identifying significant $CO_2$ emissions was probably due to emissions from rural settlements with high population densities (e.g., Sichuan Basin), which did not appear strongly in satellite nighttime lights or on the ODIAC map (Wang et al., 2013). The more diffusive distributions of MEIC and NJU were attributed to the abundance of point sources, with or without line sources and area sources proxies. Moreover, EDGAR, PKU, CHRED, MEIC and NJU all showed relatively low emissions in western China, but the emissions from ODIAC were zero due to the lack of nighttime light in that region, which tended to distribute more emissions toward strongly lit (at night) urban regions (Wang et al., 2013).

EDGAR, CHRED and MEIC all showed traffic line source emissions by inducing traffic networks in the spatial disaggregation. The line emissions (such as expressways, arterial highways) depicted a more detailed spatial distribution in CHRED than in either EDGAR or MEIC. This discrepancy could be attributed to the different road networks and corresponding weighting factors that were used by each. CHRED disaggregated emissions from the transport sector based on traffic networks and traffic flows (Cai et al., 2018). MEIC applied the traffic network from the China Digital Road-network Map (CDRM) (Zheng et al., 2017), and EDGAR traffic networks were obtained from the OpenStreetMap and OpenRailwayMap (Geofabrik, 2015). ODIAC considered point and area sources while lacking line source emissions in the spatial disaggregation, which places more emissions in populated areas than in suburbs (Oda et al., 2018). Oda and Maksyutov (2011) (Oda and Maksyutov, 2011) noted the possible utility of street lights to represent line source spatial distributions even without the associated specific traffic spatial data. The spatial distributions of traffic emissions are highly uncertain, with biases of 100% or more (Gately et al., 2015), which is largely due to mismatches between the downscaling proxies and the actual vehicle activity distribution.

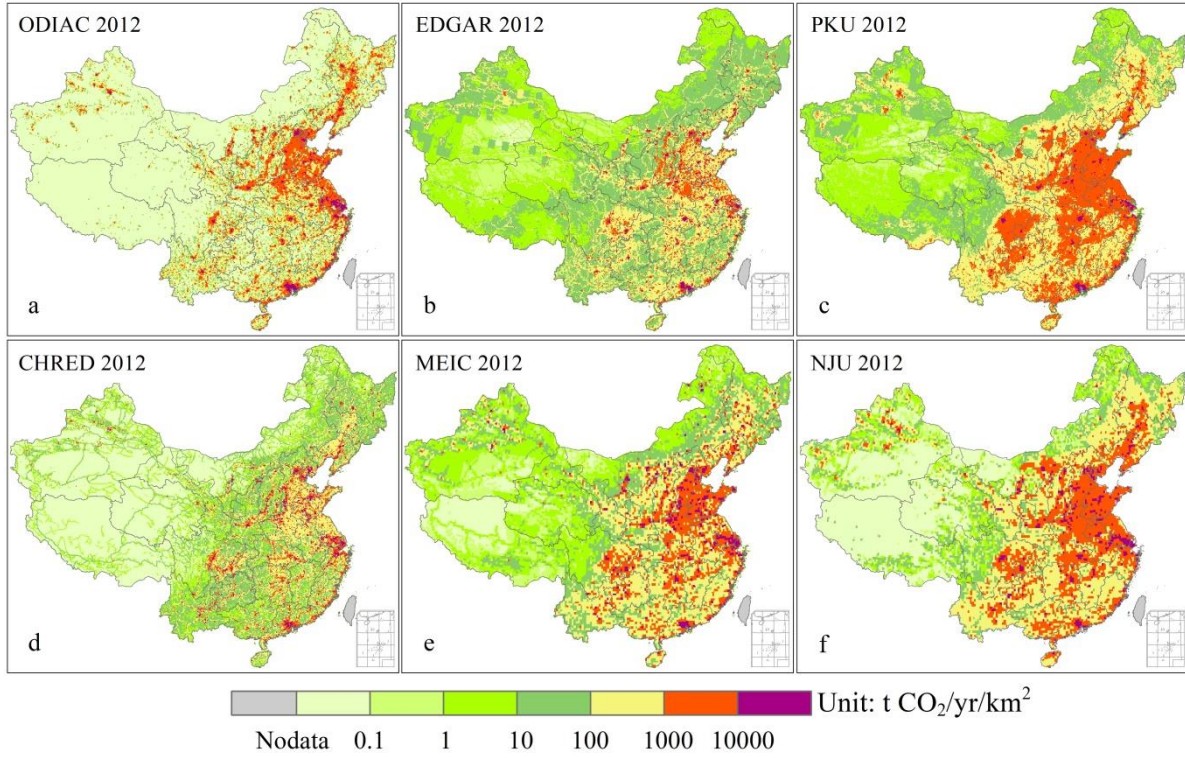

**Figure 3**. Spatial distributions of ODIAC (a), EDGAR (b), PKU (c), CHRED (d), MEIC (e) and NJU (f) at a 10 km resolution for 2012. ODIAC was aggregated from 1 km data, such that MEIC, PKU, and EDGAR were resampled from 0.25, 0.1 and 0.1 degrees.

## 4.3 $CO_2$ emissions at the provincial level

The provincial level results showed more consistency than the grid level results in terms of spatial distribution. All the products agreed that the eastern and southern provinces were high emitters (>400 Mt $CO_2$/yr, Figure 4 and S3), while the western provinces were low emitters (<200 Mt $CO_2$/yr, Figure 4 and S3). The five most emitting provinces

were Shandong, Jiangsu, Hebei, Henan and Inner Mongolia, with the emissions values ranging from 577 ±48 Mt to 820 ± 102 Mt $CO_2$ in 2012 (Figure 4). Meanwhile, the provinces located in the western area with low economic activity and population density showed low carbon emissions (<200 Mt $CO_2$, Figure 4 and S3). There is a clear discrepancy in the provincial-level emissions among the different estimates, and the mean standard deviation (SD) for the 31 provinces' emissions was 62 Mt $CO_2$ (or 20%) in 2012. A large SD (>100 Mt $CO_2$) occurred in the high-emitting provinces, such as

Shandong, Jiangsu, Inner Mongolia, Shanxi, Hebei, and Liaoning. For the Shandong Province, the inventories varied from 675-965 Mt $CO_2$/yr, with a relative SD of 12% (Figure 4 and 5), and for the other high-emitting provinces, the relative SD ranged from 12%-48%, which implied that the uncertainty could be further

reduced.

Since Because estimates based on provincial energy statistics are assumed to be more accurate than those derived from the
disaggregation of national totals using spatial proxies, we evaluated the provincial emissions of each inventory using the
provincial-based inventory mean (CHRED, MEIC, and NJU) (Figure 5). The results showed that emissions derived from the
provincial energy statistics are highly correlated, with R-values ranging from 0.99 to 1.00 and slopes ranging 0.96 to 1.04.
By In contrast, the estimates for ODIAC, EDGAR, and PKU, which used IEA national energy statistics, showed an obvious
disparity, especially in the top 5five greatestmost – emitting provinces, suggesting the large significant impact of spatial
disaggregated approaches in allocating the allocation of total emissions. The potential implication is that when doing
performing spatial disaggregation, national-data-based inventories can use provincial fractions as constraints.

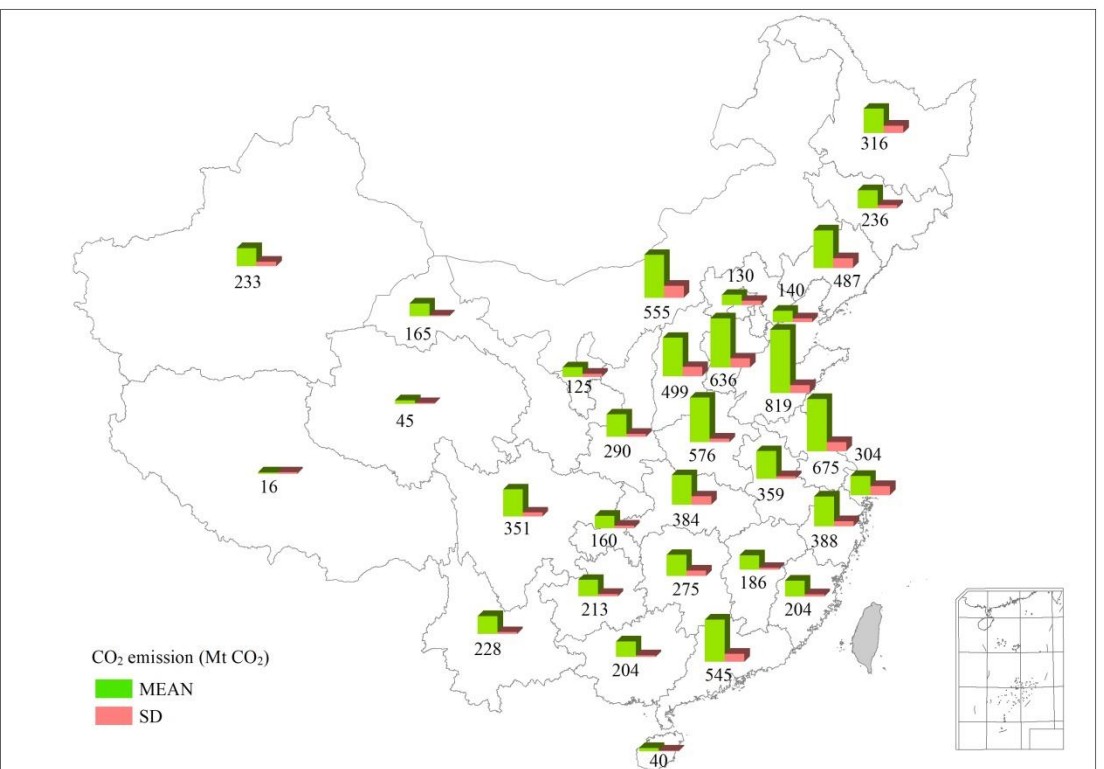

**Figure 4.** Provincial mean total emissions for ODIAC, EDGAR, PKU, CHRED, MEIC and NJU in 2012. The nNumbers refer tobeneath the green bars are the provincial total $CO_2$ emissions in Mt.

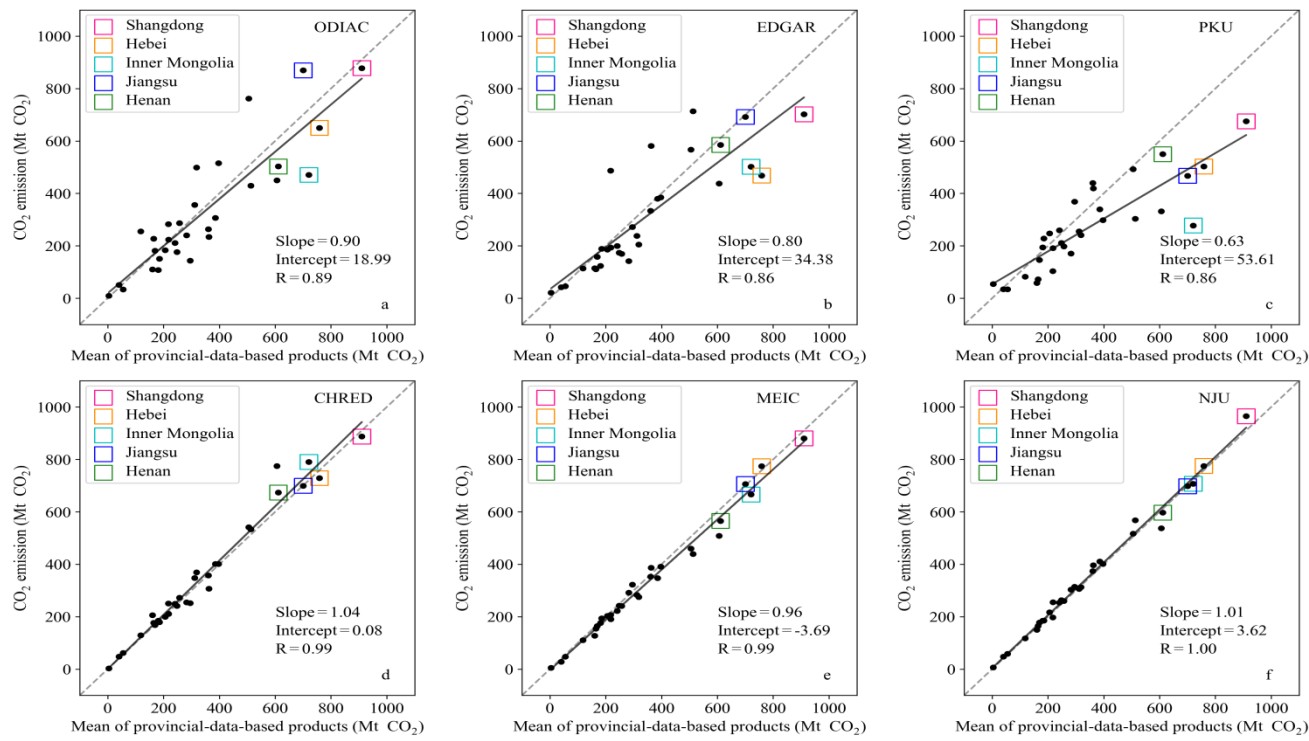

**Figure 5.** Scatter plots of ~~the~~ provincial total emissions for ODIAC, EDGAR, PKU, CHRED, MEIC and NJU in 2012 with ~~the~~ ~~top 5~~five ~~greatest~~ most emitting provinces highlighted, and the x-~~-~~axis is the mean of provincial-data-based products (CHRED, MEIC and NJU).

### 4.4 Statistics of $CO_2$ emissions at ~~the~~ grid level

To further characterize the spatial pattern of China's $CO_2$ emissions, the probability density function (PDF), cumulative

emissions, and top 5% emitting grids were analyzed to identify the spatial differences from the distribution of grid cell emissions (Figure 6). As illustrated in Figure 4a, ODIAC showed a ~~large~~ significant number of cells with zero emissions (62%) (Figure 6a), ~~with~~ medium-~~-~~emitting grids (500-50,000 t $CO_2$/km$^2$) ~~consisted~~constituted 30%~~,~~ ~~while~~ and high-~~-~~emitting grids (>50,000 t $CO_2$/km$^2$) ~~consisted~~ constituted 3%. ~~While~~ Although the low-~~-~~emissions cells (1-~~-~~500 t $CO_2$/km$^2$) were mainly located in EDGAR (58%) and CHRED (69%) (Figure 6b and d)~~,~~ and the medium-~~-~~emitting grids ~~consisted~~

constituted 30-40%, ~~while nonetheless~~ the high-~~-~~emitting grids ~~consisted~~ only amounted to 2-3%. This situation could have a ~~notable~~ significant impact on the cumulative national total emissions (Figure 6g). The frequency distribution of high-emissions grids revealed ~~the~~ differences in the~~t~~ point source data. MEIC showed the largest number of high-emitting cells (500-~~-~~500,000 t $CO_2$/km$^2$, 5% ~~compared~~ in comparison with the others, which were at 2-3%, Figure 6e) by using a high-resolution emissions database (CPED) ~~including~~ that included more power plant information (Li et al., 2017;Liu et al.,

2015a). Furthermore, ODIAC and EDGAR ~~showed a good agreement~~agreed well regarding the ~~in~~ high emissions (>~~-~~100,000 t $CO_2$/km$^2$)~~, because~~ because their point source emissions were both from the CARMA database (Table 1). Moreover,

CARMA is the only global database that tracks $CO_2$ that gathered and presented the best available estimates of $CO_2$ emissions for 50,000 power plants around the world, of which approximately 15,000 have latitude and longitude information with emissions greater than 0. The database includes approximately one-quarter of all greenhouse gas emissions. However, CARMA is no longer active (the last update was November 28, 2012), and the geolocations of power plants are not sufficiently accurate, especially in China (Byers et al., 2019;Liu et al., 2013;Wang et al., 2013;Liu et al., 2015a). Therefore, users must perform corrections themselves (Liu et al., 2013;Oda et al., 2018;Wang et al., 2013;Janssens-Maenhout et al., 2019b;Liu et al., 2015a).

As shown in the cumulative emissions plot (Figure 6g), PKU and NJU showed very similar cumulative curves, and the situation was similarly for EDGAR and CHRED. Moreover, the total emissions for EDGAR and CHRED were largely determined by a small proportion of high-emitting grids that showed a steep increase at the last stage of the cumulative curves (Figure 6g), and the top 5% emitting grids accounted for approximately 90% of the total emissions (Figure 6e), which is greater than the comparable values of 82%, 71%, 58% and 51% for ODIAC, MEIC, NJU and PKU, respectively. The emissions from PKU, MEIC and NJU were relatively evenly distributed because CHRED was mainly derived from enterprise-level point sources (Cai et al., 2018). In contrast, the emissions of PKU were the most evenly distributed, and the emissions from the top 5% emitting grids only accounted for 51% (Figure 6g) because PKU incorporated special area source survey data for Chinese rural areas from a 34,489-household energy-mix survey and a 1,670-household fuel-weighing campaign (Tao et al., 2018). Moreover, the use of a spatial disaggregation proxy based on population density also contributed to this spatial pattern. Similarly, MEIC and NJU were evenly distributed because of the same activity data from CESY, National Bureau of Statistics (Table 1).

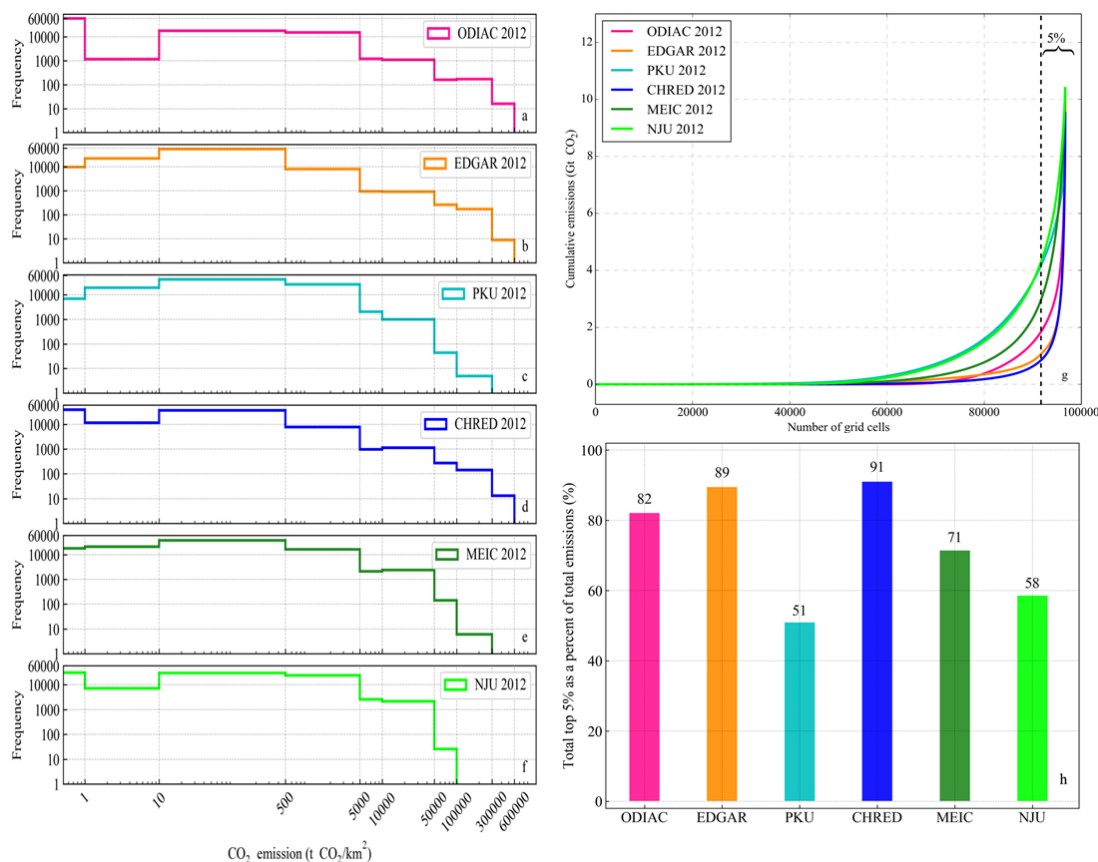

**Figure 6.** Frequency counts (a-f), cumulative emissions (g) (grids ~~were~~ are sorted from low to high)~~,~~ and the top 5% emitting grid~~s~~ plots (h) for ODIAC, EDGAR, PKU, CHRED, MEIC and NJU in 2012 at a 10-~~-~~km resolution.

To identify the locations of hotspots, ~~the~~ bubble plots (Figure S2) demonstrated the spatial distribution of high-emitting grid cells that were ~~larger~~ greater than 50 kt $CO_2/km^2$. CHRED, EDGAR and ODIAC showed ~~a~~ similar pattern~~s~~, with high-emitting grids concentrated in city clusters (e.g.~~,~~ Jing-Jin-Ji, the Yangtze River Delta~~,~~ and the Pearl River Delta) and the east~~ern~~ coast (Figure S2). EDGAR and ODIAC both derived the~~ir~~ power plant emissions from CARMA, but ODIAC was likely to ~~put~~ place more emissions than EDGAR over urbanized regions with lights, especially in the North China Plain. The emissions of CPED and CARMA were similar in China~~,~~ with a minor difference of 2%, ~~but~~ although the number~~s~~ of power plants ~~had a large difference~~ varied significantly (2320 vs. 945) (Liu et al., 2015a)~~, which~~. ~~This~~ implied that CARMA tended to allocate similar emissions to fewer plants than CPED.

## 5. Discussion

### 5.1 Activity data differences in the datasets and their effects

~~Activity~~ The activity data source~~s~~, data level and sectors ~~were~~ are the significant determin~~ants of~~ed ~~the~~ total emissions ~~largely~~. ~~As can be seen~~ As seen in Fig. 1, activity data and EF determine the total emission estimates~~,~~ and ~~then~~ affect the spatial distributions ~~through~~ by using disaggregation proxies ~~of~~ for point, line and area sources. It has been well-discussed that the

sum of ~~the~~ provincial data is ~~larger~~ greater than the national total (Guan et al., 2012;Hong et al., 2017;Liu et al., 2015b;Shan et al., 2018;Liu et al., 2013). CEADs (~~p~~Provincial) is 8-18% ~~higher~~ greater than CEADs (~~n~~National) after year 2008 (Figure

2). ~~And thus~~Thus, the province-based estimates (e.g.~~,~~ NJU and MEIC) are ~~higher~~ greater than CEADs (~~n~~National). This ~~difference~~ could be attributed to the differences in national and provincial statistical systems and artificial factors, such as the fact that some ~~of~~ provincial energy balance sheets were adjusted to ~~make to~~ achieve ~~the~~ an exact match between supply and consumption (Hong et al., 2017). For example, ~~the~~ provincial statistics ~~has~~ suffer from data inconsistency and double counting problems (Zhang et al., 2007;Guan et al., 2012). One possible way to improve th~~ese statistics is~~is to use the

provincial consumption fractions to rescale the national total consumption~~s~~ when distributing emissions to grids. Hong et al. (2017) found that the ratio of the maximum discrepancy to the mean value was 16% due to ~~the use of~~ different versions of national and provincial data in CESY. Ranges of 32-47% of $CO_2$ emissions from ~~the~~ power sector (mainly coal use) were found among ~~the~~ inventories, while for ~~the~~ transport sector (mainly liquid fuels)~~,~~ the fractions ranged from 7-9%. Apart from such differences, one peak of FFCO2 emissions was identified by most dataset~~s~~ in 2013, ~~which were~~ which was due~~largely~~

~~found to be due~~ to ~~the~~ slowing economic growth (National Bureau of Statistics, 1998–2017), changes in ~~the~~ industrial structure (Mi et al., 2017;Guan et al., 2018) and a decline in the share of coal used for energy (Qi et al., 2016)~~. S, and~~ strategies for reducing emissions could be based on such uniformed trends, while making reduction policies for provinces ~~needs~~ requires the support of provincial -energy-based datasets instead of national -energy-based ~~ones~~datasets.

Estimate~~s~~ with more sectors ~~would~~ are usually higher than those with fewer~~ sectors~~. ~~For~~ In regard to the incorporation of

different emission~~s~~ sectors, EDGAR ~~has~~ includes international aviation and bunkers (Janssens-Maenhout et al., 2017) and NJU ~~has~~ incorporates waste~~s sector~~(Liu et al., 2013) (Table S1), and thus~~,~~ both were higher than ~~the~~ others. Moreover, for MEIC_v.1.3 downloaded from ~~the~~ official website, ~~it included~~ biofuel combustion (which accounted for approximately 5.7% of the total)~~ was included, and~~; however, the version used here was specially prepared to exclude biofuel to increase ~~the database's~~ comparability of the database. ~~For another instance~~In addition, CEADs industry processes only ~~take account~~

~~of~~include cement production and was thus lower than those (e.g., NJU and EDGAR) ~~with~~ that include more processes (iron and steel, etc.) (Janssens-Maenhout et al., 2017;Shan et al., 2018;Liu et al., 2013). ~~For~~ The PKU dataset~~, it~~ used IEA energy statistics with more detailed energy ~~sub~~ subtypes. The emission~~s~~ factors ~~was~~ were based on more detailed energy ~~sub~~ subtypes with lower EFs, ~~and~~ while other inventories used ~~the~~ average~~s~~ of large groups (Table 1)~~, and~~ such that the sum of more detailed ~~sub~~ subtypes might not equal ~~to~~ the total of large groups due to ~~the~~ incomplete~~ness~~ of the statistics~~., and~~

~~t~~These factors could ~~be~~ explain the reasons for ~~its~~ the lower emissions estimate (Wang et al., 2013). A further comparison with IEA, EIA and BP estimates with only energy--related emissions also confirm~~ed~~ that estimates with more sectors would be ~~higher~~ greater than those with fewer (Figure S1).

**5.2 ~~Emission factor e~~Effects of emission factors on the total emissions**

Carbon emissions are calculated from activity data and EF~~s~~, and the uncertainty in estimates is typically reported as 5~~%~~

-10%, while the maximum difference in this study reached 33.8% (or 2.7 PgC) in 2012. One major reason for this difference is the EF used by these inventories (Table 1). The EF for raw coal ranged from 0.491 to 0.746. For example, CEADs used 0.499 tC per ton of coal based on large-sample measurements, while EDGAR used 0.713 from the default values recommended by IPCC (Janssens-Maenhout et al., 2017;Liu et al., 2015b;Shan et al., 2018), and the differences were due largely to the low quality and high ash content of Chinese coal. The variability of lignite and coal quality is quite significant. In Liu et al. (2015), the carbon content of lignite ranged from 11% to 51%, with a mean±SD of 28% ±13 (n=61). Furthermore, another study showed that the uncertainty from EFs (-16 to –24%) was significantly greater than that from activity data (-1 to 9%) (Shan et al., 2018). We recommended substituting the IPCC default coal EF with the CEADs EF. Regarding plant-level emissions from coal consumption, the collection of EFs measured at fields representing the quality and type of various coals is much needed to calibrate the large point source emissions, and we call for the inclusion of physical measurements for the calibration and validation of existing datasets (Bai et al., 2007;Dai et al., 2012;Kittner et al., 2018;Yao et al., 2019). Different fuel types contribute differently to emissions factors, i.e., for the same net heating value, natural gas emitted the least amount of carbon dioxide (61.7 kg $CO_2$/TJ energy), followed by oil (65.3 kg $CO_2$/TJ energy) and coal (94.6 kg $CO_2$/TJ energy). Similarly, one successful example for the reduction of air pollutants and $CO_2$ was that the Chinese government initiated the "project of replacement of coal with natural gas and electricity in North China" in 2016 (Zheng et al., 2018a). Moreover, the non-oxidation fraction of 8% used in Liu et al. (2015) (Liu et al., 2015b) for coal was attributable to the differences when comparing with a default non-oxidation fraction of 0%, as recommended by IPCC (2006) in EDGAR (Janssens-Maenhout et al., 2017). Moreover, the averaged qualities of coal vary with time, yet we lacked such time-series quality data on raw coal. Bottom-up inventories typically use time-invariant EFs for $CO_2$ due to the lack of information on coal heating values over time; similarly, the MEIC model also uses constant EFs of $CO_2$ (Zheng et al., 2018). Teng and Zhu (2015) recommended time-varied conversion factors from raw coal to standard coal, as well as to change the raw coal to commodity coal in energy balance statistics because the latter has relatively efficient statistics on EF.

## 5.3 Spatial distribution of point, line and area sources

### 5.3.1 Point sources in datasets and their effects on spatial distribution

Point source emissions account for a large proportion of total emissions (Hutchins et al., 2017). Power plants consumed approximately half of the total coal production in the past decade (Liu et al., 2015a). Thus, the accuracy of point sources was extremely important for improving emission estimates. ODIAC, EDGAR and PKU all distributed power plant emissions from the CARMA dataset. However, the geolocation errors in China are relatively large, and only 45% of power plants are located in the same 0.1 °×0.1 ° grid in CARMA v2.0 according to the real power plant locations that were identified by visual inspection in Google maps (Wang et al., 2013). This discrepancy is due to the

~~fact that~~because CARMA generally treats ~~the~~ city-center latitudes and longitudes as the approximate coordinates of ~~the~~ power plants (Wheeler and Ummel, 2008;Ummel, 2012).

Liu et al. (2015a) found that CARMA neglected ~~about~~ approximately 1300 small power plants in China. Thus~~.~~ CARMA allocated similar emissions to a more limited number of plants than CPED (Table S2, 720, 1706 and 2320 point sources for
ODIAC, EDGAR and MEIC, respectively), and ODIAC had fewer point sources due to ~~the~~ elimination of ~~wrong~~ incorrect geolocations. The high-emitting grids in CHRED were attributed to the 1.58 million industrial enterprises from the First China Pollution Source Census (FCPSC) that were~~,~~ used as point sources (Wang et al., 2014). Following the CARMA example, we call on the open source of large point sources for datasets and reinforce the importance that Chinese scientists ~~need to~~must adjust the locations of point sources from CARMA.

## 400 5.3.2 Effects of spatial disaggregation methods on line and area sources

Downscaling methods are widely used ~~for~~because of ~~its~~ their uniformity and simplicity ~~because of the~~due to the lack of detailed spatial data. The d~~D~~isaggregation methods used (e.g.~~,~~ nighttime light, population) by inventories ~~strongly~~ significantly affect the resulting spatial pattern. For example, ODIAC mainly use~~s~~ nighttime light from satellite images to distribute emissions. Thus~~,~~ the hotspots concentrate~~d~~ more ~~in~~ strong~~ly in high~~ nighttime light regions. However, ~~using~~ the
405 use of remote sensing data tend~~s~~ed to underestimate industrial and transportation emissions (Ghosh et al., 2010). For instance, coal-fired power plants do not emit strong lights and may be far ~~away~~ from cities ~~by~~because transmission lines are used. Electricity generation and use ~~are~~ usually ~~happened at~~occur in different ~~places~~locations, and stronger night-time light does not always ~~mean~~ indicate higher $CO_2$ emissions (Cai et al., 2018;Doll et al., 2000). Furthermore, night-time lights ~~would~~ ignore some other main fossil fuel emissions~~,~~ such as household cooking with coal. The good correlation between night-time
light and $CO_2$ emissions is usually on a larger scale basis (national or continental) (Oda et al., 2010;Raupach et al., 2010), while this relationship ~~would~~ fail~~s~~ in populated or industrialized rural areas.

Transport networks are also used in several inventories for spatial disaggregation. EDGAR and CHRED both showed clear transport emissions~~,~~ especially in western China. EDGAR use~~s~~d three road types and their corresponding weighting factors to disaggregate line source emissions. CHRED use~~s~~d national traffic networks and their flows to distribute traffic emissions
(Cai et al., 2018;Cai et al., 2012). It is easier to obtain ~~the~~ traffic networks but rather difficult to ~~get~~obtain ~~the~~ traffic flows and vehicle kilometer~~s~~ travelled (VKT) data, and thus~~,~~ the weighting factors method ~~are~~is ~~much~~ significantly easier to apply. Population is widely used in spatial disaggregation (Andres et al., 2014;Andres et al., 2016;Janssens-Maenhout et al., 2017). ~~The~~ CDIAC emission~~s~~ maps originally used ~~a~~ static population data to distribute emissions ~~and~~but have recently ~~have~~ changed to a temporally varying population proxy, which has largely reduced ~~the~~ uncertainty. However, the unified algorithm
for spatial disaggregation~~,~~ such as the population density approach~~,~~ ~~has~~ encounters difficulties in depicting the uneven development of rural and urban areas, and instead, it usually use~~s~~ interpolation for a limited number of base years and does not truly vary across years at high spatial resolution (Andres et al., 2014). Furthermore, downscaling approaches may

introduce approximately 50% error per pixel, which are spatially correlated (Rayner et al., 2010), ~~and this problem needs to~~a problem ~~which~~that must be considered in future studies.

Moreover, big cities have virtually eliminate~~d the~~d use of coal (Guan et al., 2018;Zheng et al., 2018a), while in rural areas. the use of coal has ~~even~~often increased (Meng et al., 2019). For example, a national survey showed that China's rural residential coal consumption fractions for heating increased from 19.2% to 27.2% (Tao et al., 2018). These transitions ~~has~~ have impacts on the spatial distribution of both $CO_2$ and air pollutants. ~~And~~In addition, the high-~~ ~~resolution $CO_2$ emissions ~~have~~can serve as a potential proxy for fossil fuel emissions (Wang et al., 2013)~~;~~. thus, further improvements ~~on~~to spatial

disaggregation should consider these transitions and the survey~~ed~~ data.

*Data availability.* The data~~—~~sets of ODIAC, EDGAR, PKU and CEADs are freely available from

435 http://db.cger.nies.go.jp/dataset/ODIAC/, http://edgar.jrc.ec.europa.eu/overview.php?v=432_GHG&SECURE=123, http://inventory.pku.edu.cn/download/download.html and http://www.ceads.net/~~ respectively~~, respectively. ~~And~~ CHRED, MEIC and NJU are available from the data developers upon request.

*Author contributions.* PFH and NZ conceived and designed the study. PFH and XHL collected and analyzed the data

sets. PFH, XHL, NZ and TO ~~led the paper writing~~wrote the paper, with contributions from all the coauthors.

*Competing interests.* The authors declare that they have no conflict of interest.

*Acknowledgments.* This work was supported by the National Key R&D Program of China (No. 2017YFB0504000). We

thank Dr. Bofeng Cai from the Chinese Academy for Environmental Planning for kindly providing the CHRED data and his suggestions for improving the manuscript.

*Supporting Information.* Data and methodology descriptions o~~f~~n the nine~~9~~ datasets and supplementary figures on emission estimates

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

SCIO, T. S. C. I. O. o. C.: Enhanced Actions on Climate Change: China's Intended Nationally Determined
Contributions,
http://www.scio.gov.cn/xwfbh/xwbfbh/wqfbh/33978/35364/xgzc35370/Document/1514539/151453
9.htm, 2015.

Shan, Y., Liu, J., Liu, Z., Xu, X., Shao, S., Wang, P., and Guan, D.: New provincial CO2 emission
inventories in China based on apparent energy consumption data and updated emission factors,
Applied Energy, 184, 2016.

Shan, Y., Guan, D., Zheng, H., Ou, J., Li, Y., Meng, J., Mi, Z., Liu, Z., and Zhang, Q.: China CO2 emission
accounts 1997-2015, Scientific Data, 5, 170201, 2018.

Tao, S., Ru, M. Y., Du, W., Zhu, X., Zhong, Q. R., Li, B. G., Shen, G. F., Pan, X. L., Meng, W. J., Chen, Y. L.,
Shen, H. Z., Lin, N., Su, S., Zhuo, S. J., Huang, T. B., Xu, Y., Yun, X., Liu, J. F., Wang, X. L., Liu, W. X., Cheng,
H. F., and Zhu, D. Q.: Quantifying the rural residential energy transition in China from 1992 to 2012
through a representative national survey, Nature Energy, 3, 567-573, 10.1038/s41560-018-0158-4,
2018.

Teng, F., and Zhu, S.: Which estimation is more accurate? A technical comments on Nature Paper by
Liu et al on overestimation of China's emission, Science & Technology Review, 33, 112-116, 2015.

Ummel, K.: Carma Revisited: An Updated Database of Carbon Dioxide Emissions from Power Plants
Worldwide, Working Papers, 2012.

Wang, J., Cai, B., Zhang, L., Cao, D., Liu, L., Zhou, Y., Zhang, Z., and Xue, W.: High resolution carbon
dioxide emission gridded data for China derived from point sources, Environmental Science &
Technology, 48, 7085-7093, 2014.

Wang, M., and Cai, B.: A two-level comparison of CO2 emission data in China: Evidence from three
gridded    data    sources,    Journal    of    Cleaner    Production,    148,    194-201,

https://doi.org/10.1016/j.jclepro.2017.02.003, 2017.

Wang, R., Tao, S., Ciais, P., Shen, H. Z., Huang, Y., Chen, H., Shen, G. F., Wang, B., Li, W., Zhang, Y. Y., Lu, Y., Zhu, D., Chen, Y. C., Liu, X. P., Wang, W. T., Wang, X. L., Liu, W. X., Li, B. G., and Piao, S. L.: High-resolution mapping of combustion processes and implications for CO2 emissions, Atmos. Chem. Phys., 13, 5189-5203, https://doi.org/5110.5194/acp-5113-5189-2013, 2013.

Wheeler, D., and Ummel, K.: Calculating CARMA: Global Estimation of CO2 Emissions from the Power Sector, Working Papers, 2008.

Yao, B., Cai, B., Kou, F., Yang, Y., Chen, X., Wong, D. S., Liu, L., Fang, S., Liu, H., Wang, H., Zhang, L., Li, J., and Kuang, G.: Estimating direct CO2 and CO emission factors for industrial rare earth metal electrolysis, Resources, Conservation and Recycling, 145, 261-267, https://doi.org/10.1016/j.resconrec.2019.02.019, 2019.

Zeng, N., Ding, Y., Pan, J., Wang, H., and Gregg, J.: Climate Change--the Chinese Challenge, Science, 319, 730-731, 10.1126/science.1153368, 2008.

Zhang, Q., Streets, D. G., He, K., and Klimont, Z.: Major components of China's anthropogenic primary particulate emissions, Environmental Research Letters, 2, 045027, 2007.

Zheng, B., Zhang, Q., Tong, D., Chen, C., Hong, C., Li, M., Geng, G., Lei, Y., Huo, H., and He, K.: Resolution dependence of uncertainties in gridded emission inventories: a case study in Hebei, China, Atmos. Chem. Phys., 17, https://doi.org/10.5194/acp-17-921-2017, 2017.

Zheng, B., Tong, D., Li, M., Liu, F., Hong, C., Geng, G., Li, H., Li, X., Peng, L., Qi, J., Yan, L., Zhang, Y., Zhao, H., Zheng, Y., He, K., and Zhang, Q.: Trends in China's anthropogenic emissions since 2010 as the consequence of clean air actions, Atmos. Chem. Phys., 18, 14095-14111, https://doi.org/14010.15194/acp-14018-14095-12018, , 2018a.

Zheng, B., Zhang, Q., Davis, S. J., Ciais, P., Hong, C., Li, M., Liu, F., Tong, D., Li, H., and He, K.: Infrastructure Shapes Differences in the Carbon Intensities of Chinese Cities, Environmental Science & Technology, 52, 6032-6041, 10.1021/acs.est.7b05654, 2018b.