# Peer review of "Evaluating China's fossil-fuel CO$_2$ emissions from a comprehensive dataset of nine inventories"

_Atmospheric Chemistry and Physics, 2019_

## Referee Comment (RC1) · Anonymous Referee #1 · 28 Apr 2020

Reducing uncertainty in China's CO2 emissions and understanding its trends is very relevant of course. The presented attempt of a thorough comparison of nine inventories can be useful but in the current form I find it unconvincing. Some sections are not written clearly and do not present clear findings or conclusions. I find the balance between disusing the CO2 emission sources and strengths and the spatial distribution of CO2 is not right, the latter receives most of the attention while I think it should be the other way around, or in fact the discussion of emission and trends (section 4.1 is only 1 page) should be expanded. I believe, the paper needs a major revision but most of that should be deeper analysis and better characterization/discussion of reasons for differences and what does it mean for the future, ie., how can we do better. Still, I believe this work shall be published and with all the material collected and already

evaluated to some extent, the manuscript can be revised successfully.

Here are more specific comments that shall help to understand why I made the above statement.

Abstract: Line 31: Bearing in mind uncertainties, using 'about/around' rather than a precise 28% might be more appropriate. Line 37: Suggest adding a unit for the emission factors. Additionally (and this is something more important for the section discussing emission factors), there are some good reasons for variability in $CO_2$ EF for coal as well as change over time (this is something that is not discussed enough in the paper) and so the authors could potentially revisit this statement later after revision.

Introduction: I recommend a closer look at the whole introduction and consider rewriting it. I find it lack structure and order; it contains lots of information and references but all of it appears to be arranged in a bit chaotic way. A clear separation of discussion of total emissions and trends from spatial distribution would help for example, now these are mixed in different paragraphs (see for example 2nd para).

Also, please check Reference style as in the text references use of 'single names' as authors while many of those are papers with many authors as given in the Reference section. The $CO_2$ emission inventories are uncertain everywhere, highlighting why Chinese are possible more uncertain and why it matters would be important.

Line 49-50: suggest adding a reference to IPCC AR5 too

2.Emission data As shown in Table 1, the evaluated inventories are covering various periods but overlap. I'd expect that after reading this section (line 107-113) one would know for which years the evaluation will be performed. In fact, even in the method section (3), this is not evident.

I see that in the SI, there is an extended version of Table 1. I was wondering if adding a row with EFs for cement industry across inventories would be also useful.

I think it would be useful to add a short paragraph explaining why it is important to

evaluate spatial distribution of CO2 emissions. It is certainly obvious for the authors and many but not for all. Not sure if this is the best place but (could be also in the introduction or method).

3. Methods I am struggling a little to understand the significance of the Figure 1 as the concept for the evaluation method. The figure does not show anything beyond obvious and sources of data or sectors are listed in further text anyway. In general, I find this whole section is not written very well or informative yet; in fact, beyond the 2nd para where there is some information about spatial analysis I do not see here much of a concept or method explained. I think, this needs further work and clear statement why and how certain things are done and why priority is given to X or Y. Additionally, some of the assumptions about the considered sectors for comparison could be briefly discussed here as inventories do not have the same sources included [some of that is mentioned in the Discussion section but I believe it should be already brought in here] and for a comparison it would be sensible to assure apples are compared to apples as much as possible.

Line 125: Is "nearest neighbor algorithm' a standard used name and most will be familiar with it?

4. Results I think section 4.1 needs some clearer writing and add discussion as to why the range and uncertainties grow with time. I find the discussion of total (and also sectoral) emissions and trends deserves a lot more space and I find it more important than spatial distribution.

Line 146: ref to point 1); the EFs were the same for all sectors? Are these country averages? Were they changing over time in these or other inventories? I think it might be useful to add this discussion.

Line 149: What are "differences in emission definitions"? Do you mean sources? If so, then it might be important to try to bring it to a common denominator and if not possible then say why and what implications it has rather than saying they are different.

Line 153: MEIC EF lower than EDGAR? Both average over all sectors for coal, or all fuels? In what units? How did change over time if inventories consider this (I think MEIC does).

Line 155: "minor difference" Is this good? Sensible? Or the match seems fine but maybe for wrong reason? As mentioned earlier the whole 4.1 misses actual discussion.

Line 184: "could be attributed"; If information is available and assume it is, then maybe one shall be more certain about it and say "is attributed" stating that this was identified as a reason.

Line 188: editorial "ODIAC was lack"

Fig3. The scale/ranges selected are a bit odd, changing x10/x2.5/x2/x5/x2 and so it makes interpretation of differences a bit more challenging.

Line 200-2002: Why is this important for cumulative total? I assumed that the spatial distribution comes after emissions are calculated?

Line 206: I believe somewhere in Discussion section there is mention of issues/completeness of CARMA (and a reference to the paper evaluating it) but it would be useful to mention this also here I think

Line 241-244: Isn't it obvious? Anything different would be strange, wouldn't it?

Fig 5: Editorial: The numbers are actually not always 'under' so it might be better to say that the numbers simply refer to the green bars

5. Discussion Suggest to revisit the whole 5.1 to improve clarity. I am struggling to understand several statements here.

Line 256: "Artificial factors" – what is meant here?

Line 257-259: I have difficulty to understand what is suggested here.

Line 265: First sentence; what does it mean? It hints that possibly the comparison is

not really addressing the same sources and that in some inventories some are missing? If so, then I think this should be mentioned much earlier and then statements about which specific sectors are of concern and if the other (common) sectors compare reasonably.

Section 5.2 could benefit from additional discussion of: - % of $CO_2$ from coal use vs. cement production vs. liquid fuels (transport, etc) - Differences between coals used in different sectors; where such info exists and how important it could be - Change in EFs (especially for coal) over time owing to potentially declining or improving fuel quality in specific sectors

Section 5.4 – I think the title should really explicitly refer to 'area sources and line sources'. In fact, I thought that one can have one section 5.3 for "spatial distributon' and then sub sections on point sources and area sources.

Table S1: I find this table very difficult to read. One should consider reformatting and to show included source-sectors in each inventory I'd suggest to make a row for each sector and then 'tick' the ones that are covered in specific inventory. It would make reading of the table much easier.

---

## Referee Comment (RC2) · Anonymous Referee #2 · 19 Jun 2020

This is a potentially very interesting paper but the current version is poorly organized and inadequately explained. It needs serious reorganization to tell a direct and clear story. Most of the graphics are quite adequate but need a bit more explanation. Figures 4a, 4b, etc need a lot more explanation.

The problems start early. This is a comparison of 9 datasets but sentence number 3 seems to accept values from one dataset (Le Quere et al.) – but this dataset does not appear to be part of the comparison. Table 1 lists the properties of the datasets to be compared, but there are only 7 in the table. Sentence number 2 of paragraph 2 jumps abruptly and without explanation from Chinese emissions estimates to global gridded emissions datasets. Line 58 introduces the CDIAC dataset, which also turns out to be not part of the comparison. In line 100 datasets from EIA, IEA, and BP are introduced,

but also apparently not used in the comparison. There is no consistent story line on what is being compared and why, on the fact that comparison will be made at the national, provincial, and grid bases. Table 1, text line 110, and Figure 2 all seem to say that CHRED exists only for 2007, but it appears elsewhere, for example Figure 4, with data for 2012?

Interesting but ad hoc statements appear throughout the text. Line 102 says that one of the purposes of the study was to identify "spatiotemporal differences" but there is no further mention of temporal differences. CARMA enters the discussion in line 109, without definition or citation. EF enters the discussion in line 88 but if it is defined it is lost in a sea of acronyms. Biomass burning appears in line 118 but there is no mention, until the closing discussion, on how it is used. I did not find enough discussion of Figure 1 to make it useful. IF FCPSC was defined I missed it.

Line 139 says "both are 21%". But does not say % of what. This problem appears elsewhere in the text as well. In line 299, "same" as what? Text around lines 145 to 155 is so poorly organized that it is hard to follow. Page 10 is rambling and disconnected.

On line 225 do I understand that total emissions from large point sources are approximately the same even though one data set has 2320 points and the other 945? How does this fit in with the 720, 1706, and 2320 in line 303?

I could list many additional problems of organization and flow of the text.

There is much here that appears to be interesting. The paper needs a major reorganization and significant increase in explanations of what was done and why and what we learn from it.

---

## Author Comment (AC1) · 30 Jul 2020

Responses to reviewer 1 Dear Editor and Reviewer, We thank you for your letter and for the reviewer's constructive comments concerning our manuscript. Those comments are all very valuable and helpful for revising and improving our paper. We have studied your comments carefully and have made corrections accordingly, which we hope to have addressed your concerns. Revised parts are marked in track change mode in the paper. The main corrections in the paper and the responses to the reviewer's comments are as follows.

Anonymous Referee #1 Reducing uncertainty in China's CO2 emissions and understanding its trends is very relevant of course. The

[Figure]

presented attempt of a thorough comparison of nine inventories can be useful but in the current form I find it unconvincing. Some sections are not written clearly and do not present clear findings or conclusions. I find the balance between disusing the CO2 emission sources and strengths and the spatial distribution of CO2 is not right, the latter receives most of the attention while I think it should be the other way around, or in fact the discussion of emission and trends (section 4.1 is only 1 page) should be expanded. I believe, the paper needs a major revision but most of that should be deeper analysis and better characterization/discussion of reasons for differences and what does it mean for the future, ie., how can we do better. Still, I believe this work shall be published and with all the material collected and already evaluated to some extent, the manuscript can be revised successfully. Here are more specific comments that shall help to understand why I made the above statement. Response: We thank you for understanding the value of this paper. And we revised the MS as you have suggested. Abstract: Line 31: Bearing in mind uncertainties, using 'about/around' rather than a precise 28% might be more appropriate. Response: Thank you. Revised accordingly. Line 37: Suggest adding a unit for the emission factors. Additionally (and this is something more important for the section discussing emission factors), there are some good reasons for variability in CO2 EF for coal as well as change over time (this is something that is not discussed enough in the paper) and so the authors could potentially revisit this statement later after revision. Response: Thank you. Added unit for coal EF (t C per t of coal). Indeed the EF for coal varied with time due to the changing coal quality, we added this in the discussion (lines 413-417). Averaged coal qualities are varying with time, yet we lacked such time-series quality data on raw coal. Bottom-up inventories typically use time-invariant EFs for CO2 due to the lack of information on coal heating values over time and the MEIC model also uses constant EFs of CO2 (Zheng et al., 2018). Teng and Zhu (2015) recommended time varied conversion factors from raw coal to standard coal, and change the raw coal to commodity coal in energy balance statistics since the latter has relatively efficient statistics on EF. Moreover, Liu et al. (2015b) considered the EFs and fractions of imported coal

and local productions and thus the weighted value reflected coal quality varying to a certain degree. Introduction: I recommend a closer look at the whole introduction and consider rewriting it. I find it lack structure and order; it contains lots of information and references but all of it appears to be arranged in a bit chaotic way. A clear separation of discussion of total emissions and trends from spatial distribution would help for example, now these are mixed in different paragraphs (see for example 2nd para). Also, please check Reference style as in the text references use of 'single names' as authors while many of those are papers with many authors as given in the Reference section. The $CO_2$ emission inventories are uncertain everywhere, highlighting why Chinese are possible more uncertain and why it matters would be important. Response: Thank you so much for your constructive suggestions. We rearranged it as you suggested. We separated total emissions and spatial disaggregation through rewriting the 2nd and 3rd paragraphs. We arranged the introduction from a general background of China's fossil fuel $CO_2$ emissions, and then to the total estimates and spatial proxies, followed by the local inventories developed within China using more detailed provincial activity data and local optimized emission factors. Finally as you have suggested, we pointed out the importance of this study: Why Chinese are possible more uncertain and why it is important. Thank you for the careful review. We checked and corrected the Ref styles, and the original wrong format was caused by incorrect comma used in EndNote. Line 49-50: suggest adding a reference to IPCC AR5 too Response: Thank you. We added this reference. 2. Emission data As shown in Table 1, the evaluated inventories are covering various periods but overlap. I'd expect that after reading this section (line 107-113) one would know for which years the evaluation will be performed. In fact, even in the method section (3), this is not evident. Response: Thank you for this advice. We added it (year 2012) in lines 160-162. I see that in the SI, there is an extended version of Table 1. I was wondering if adding a row with EFs for cement industry across inventories would be also useful. Response: Thank you for this suggestion. We added EFs for cement productions. I think it would be useful to add a short paragraph explaining why it is important to evaluate spatial distribution of $CO_2$ emissions. It is certainly

obvious for the authors and many but not for all. Not sure if this is the best place but (could be also in the introduction or method). Response: Thank you. We added such explanations in the introduction part (lines 73-77). The gridded products provide basic understanding of where emissions come from and provide key inputs for transport and data assimilation models. Furthermore, policy makers can use this information for emissions reductions and environmental monitoring can use it for instruments deployment. 3. Methods I am struggling a little to understand the significance of the Figure 1 as the concept for the evaluation method. The figure does not show anything beyond obvious and sources of data or sectors are listed in further text anyway. In general, I find this whole section is not written very well or informative yet; in fact, beyond the 2nd para where there is some information about spatial analysis I do not see here much of a concept or method explained. I think, this needs further work and clear statement why and how certain things are done and why priority is given to X or Y. Additionally, some of the assumptions about the considered sectors for comparison could be briefly discussed here as inventories do not have the same sources included [some of that is mentioned in the Discussion section but I believe it should be already brought in here]and for a comparison it would be sensible to assure apples are compared to apples as much as possible. Response: Thank you for this suggestion. We added more information in lines 146-149. Actually Fig. 1 only depicts the conceptual procedure in total emissions estimates and how gridded maps are produced for all inventories for a broad range of readers, who may be not specialized or familiar with inventories, thus it is important to know the differences in activity data, EF and spatial proxy data and spatial disaggregation methods they used, to further understand the differences among inventories in total emissions estimates and spatial characteristics. We totally agree with you on this point that assuring apples are compared to apples as much as possible. And we followed this principle to include only the fossil fuel $CO_2$ (FFCO2) and industry processes associated $CO_2$ emissions. We excluded not so comparable inventories, such as BP and IEA, which only considered FFCO2. Line 125: Is "nearest neighbor algorithm' a standard used name and most will be familiar with it? Response:

Thank you, and we added an explanation on this term. 4. Results I think section 4.1 needs some clearer writing and add discussion as to why the range and uncertainties grow with time. I find the discussion of total (and also sectoral) emissions and trends deserves a lot more space and I find it more important than spatial distribution. Response: Thank you. We added discussions in lines 193-194 explaining the range and uncertainties grow with time. Although the range increased with time, the relative difference remains at around 21%, indicating the systematical differences such as EFs remains stable. And we added more discussions on EF and emissions sectors in lines 178-187, 335, and 346. Line 146: ref to point 1); the EFs were the same for all sectors? Are these country averages? Were they changing over time in these or other inventories? I think it might be useful to add this discussion. Response: Thank you. We added this discussion in lines 178-182. The EFs were different for different fossil fuel types and cement production (Table S2). And they are from either IPCC default values or local optimized values from different sources. They generally do not change over time in these inventories although they should due to the unavailability of EFs over time (Teng and Zhu, 2015;Zheng et al., 2018). Line 149: What are "differences in emission definitions"? Do you mean sources? If so, then it might be important to try to bring it to a common denominator and if not possible then say why and what implications it has rather than saying they are different. Response: Yes, here we mean emission sources or sectors. We agree with you and added in lines 185-187. Although we tried to make these datasets as comparable as possible, there are still minor differences in emission sources (sectors). For example, EDGAR contains abundant industry processes emissions while CEADs only considered cement production. Line 153: MEIC EF lower than EDGAR? Both average over all sectors for coal, or all fuels? In what units? How did change over time if inventories consider this (I think MEIC does). Response: Indeed, MEIC used lower EFs from CEADs (Zheng et al., 2018) than EDGAR for coal and cement productions, while for oil and gas the EF are very close (Table S1). For coal, the EF for MEIC is 0.499 tC per t coal, while EDGAR used IPCC default values, for coal it is 0.713 tC per t coal. Bottom-up inventories typically use time-invariant

EFs for CO2 due to the lack of information on coal heating values over time and the MEIC model also uses constant EFs of CO2 (Zheng et al., 2018). Line 155: "minor difference" Is this good? Sensible? Or the match seems fine but maybe for wrong reason? As mentioned earlier the whole 4.1 misses actual discussion. Response: The "minor differences in magnitude" may be misleading, so we changed this into "small differences in magnitude of total emissions estimates". As explained above, the relative difference remains around 21%, indicating the potential systematical differences in EFs, which do not change over time. Line 184: "could be attributed"; If information is available and assume it is, then maybe one shall be more certain about it and say "is attributed" stating that this was identified as a reason. Response: We agree with you. We changed it to "is attributed". Line 188: editorial "ODIAC was lack" Response: Sorry that we cannot quite understand the comment on Line 188: editorial "ODIAC was lack", and we added more background information in line 228. Here we mean that ODIAC included point sources and area sources and do not have line sources in spatial disaggregation. Fig3. The scale/ranges selected are a bit odd, changing x10/x2.5/x2/x5/x2 and so it makes interpretation of differences a bit more challenging. Response: Thank you for the question. We revised the ranges in Fig.3 to x10. In the old versions, we tried several schemes and used those ranges to fully reflect the differences of inventories to avoid potential saturation of colors. Line 200-2002: Why is this important for cumulative total? I assumed that the spatial distribution comes after emissions are calculated? Response: Thank you for the question. The spatial distribution indeed comes after total emissions are calculated. Cumulative total is important in understanding the spatial distributions, potential use for assignment of responsibilities in emissions reductions, and also for modeling studies that focus on spatial distributions of carbon dioxide sources and sinks. Line 206: I believe somewhere in Discussion section there is mention of issues/completeness of CARMA (and a reference to the paper evaluating it) but it would be useful to mention this also here I think Response: Thank you for this good suggestion. We added completeness and issues of CARMA in lines 305-311. CARMA is the only global database for tracking CO2 that gathered and presented the

best available estimates of CO2 emissions for 50,000 power plants around the world, of which around 15, 000 have latitude and longitude information with emissions larger than 0. The database is responsible for about one-quarter of all greenhouse gas emissions. However, CARMA is no longer active (the last update was November 28, 2012), and the geolocations of power plants are not accurate enough, especially in China (Byers et al., 2019;Liu et al., 2013;Wang et al., 2013;Liu et al., 2015a). Therefore users have to do corrections themselves (Liu et al., 2013;Oda et al., 2018;Wang et al., 2013;Janssens-Maenhout et al., 2019;Liu et al., 2015a). Line 241-244: Isn't it obvious? Anything different would be strange, wouldn't it? Response: Indeed, you are right, and this is assumed to be so and proved to be. In presenting this, we also want to indicate that when doing spatial disaggregation, national-data-based inventories can use provincial fractions as constraints. Since national inventories do not directly include provincial information, they can use the weights from provincial data based inventories to rescale and redistribute the national total estimates. Fig 5: Editorial: The numbers are actually not always 'under' so it might be better to say that the numbers simply refer to the green bars Response: Thank you for the careful review. Revised accordingly. 5. Discussion Suggest to revisit the whole 5.1 to improve clarity. I am struggling to understand several statements here. Response: Thank you. We added more explanations and discussions around lines 335-347. Line 256: "Artificial factors" – what is meant here? Response: Added descriptions in lines 341-342. Here"Artificial factors" means data may be adjusted artificially to meet certain goals. For example, Hong et al. (2017) pointed out that some provinces had zero statistical difference; that is the supply data matches the consumption data exactly, which may indicate that some provincial data were adjusted to achieve the exact match. Moreover, the energy revisions in 2005 and 2010 have adjusted the total national energy use with special attention to the annual coal consumption (Guan et al., 2012), after the second economic census it was found to bring the country closer to achieving its energy conservation targets (Aden, 2010). Line 257-259: I have difficulty to understand what is suggested here. Response: Here we mean that the fractions of provincial emissions in province-data-based inventories

can serve as regional constraints in spatial disaggregation of national-data-based inventories. Since national inventories do not directly include provincial activity data information, they can use the weights from provincial data based inventories to rescale and redistribute the national total estimates. Line 265: First sentence; what does it mean? It hints that possibly the comparison is not really addressing the same sources and that in some inventories some are missing? If so, then I think this should be mentioned much earlier and then statements about which specific sectors are of concern and if the other (common) sectors compare reasonably. Response: Yes, you are right, and we added explanations in lines 185-187. Although we tried to make the inventories as comparable as possible, some minor sources/sectors are different among inventories (see table S1 emission sectors for detail), and most datasets only provide total estimates or major sectors, and do not provide such detailed sub-sectors data. Section 5.2 could benefit from additional discussion of: - % of $CO_2$ from coal use vs. cement production vs. liquid fuels (transport, etc) - Differences between coals used in different sectors; where such info exists and how important it could be - Change in EFs (especially for coal) over time owing to potentially declining or improving fuel quality in specific sectors Response: Thank you for the good suggestions. And we added such discussions in lines 346-347. Such results (see below figure) are already prepared in another sectorial comparison paper.

Fractions of sectoral emissions in inventories. Section 5.4 – I think the title should really explicitly refer to 'area sources and line sources'. In fact, I thought that one can have one section 5.3 for "spatial distributon' and then sub sections on point sources and area sources. Response: We agree with you on this good idea. We combined 5.3 and 5.4 and make point, line and area sources as sub sections. Table S1: I find this table very difficult to read. One should consider reformatting and to show included source-sectors in each inventory I'd suggest to make a row for each sector and then 'tick' the ones that are covered in specific inventory. It would make reading of the table much easier. Response: Thank you for this good suggestion, and we revised accordingly.

References: Initial Assessment of NBS Energy Data Revisions, 2010. Byers, L., Friedrich, J., Hennig, R., Kressig, A., Li, X., McCormick, C., and Malaguzzi, V. L.: A Global Database of Power Plants, in, World Resources Institute. Available online at www.wri.org/publication/global-database-power-plants., Washington, DC, 2019. Guan, D., Liu, Z., Geng, Y., Lindner, S., and Hubacek, K.: The gigatonne gap in China's carbon dioxide inventories, Nature Climate Change, 2, 672-675, 10.1038/nclimate1560, 2012. Hong, C., Zhang, Q., He, K., Guan, D., Li, M., Liu, F., and Zheng, B.: Variations of China's emission estimates: response to uncertainties in energy statistics, Atmos. Chem. Phys., 17, 1227-1239, https://doi.org/1210.5194/acp-1217-1227-2017, 2017. Janssens-Maenhout, G., Crippa, M., Guizzardi, D., Muntean, M., Schaaf, E., Dentener, F., Bergamaschi, P., Pagliari, V., Olivier, J. G. J., Peters, J. A. H. W., van Aardenne, J. A., Monni, S., Doering, U., Petrescu, A. M. R., Solazzo, E., and Oreggioni, G. D.: EDGAR v4.3.2 Global Atlas of the three major greenhouse gas emissions for the period 1970–2012, Earth Syst. Sci. Data, 11, 959-1002, 10.5194/essd-11-959-2019, 2019. Liu, F., Zhang, Q., Tong, D., Zheng, B., Li, M., Huo, H., and He, K. B.: High-resolution inventory of technologies, activities, and emissions of coal-fired power plants in China from 1990 to 2010, Atmos. Chem. Phys., 15, 13299-13317, 2015a. Liu, M., Wang, H., Oda, T., Zhao, Y., Yang, X., Zang, R., Zang, B., Bi, J., and Chen, J.: Refined estimate of China's $CO_2$ emissions in spatiotemporal distributions, Atmos. Chem. Phys., 13, 10873-10882, https://doi.org/10810.15194/acp-10813-10873-12013, 2013. Liu, Z., Guan, D., Wei, W., Davis, S. J., Ciais, P., Bai, J., Peng, S., Zhang, Q., Hubacek, K., Marland, G., Andres, R. J., Crawford-Brown, D., Lin, J., Zhao, H., Hong, C., Boden, T. A., Feng, K., Peters, G. P., Xi, F., Liu, J., Li, Y., Zhao, Y., Zeng, N., and He, K.: Reduced carbon emission estimates from fossil fuel combustion and cement production in China, Nature, 524, 335, 10.1038/nature14677 https://www.nature.com/articles/nature14677#supplementary-information, 2015b. Oda, T., Maksyutov, S., and Andres, R. J.: The Open-source Data Inventory for Anthropogenic $CO_2$, version 2016 (ODIAC2016): a global monthly fossil fuel $CO_2$ gridded emissions data product for tracer transport simulations and surface flux inversions,

Earth Syst. Sci. Data, 10, 87-107, https://doi.org/110.5194/essd-5110-5187-2018, 2018. Teng, F., and Zhu, S.: Which estimation is more accurate? A technical comments on Nature Paper by Liu et al on overestimation of China's emission, Science & Technology Review, 33, 112-116, 2015. Wang, R., Tao, S., Ciais, P., Shen, H. Z., Huang, Y., Chen, H., Shen, G. F., Wang, B., Li, W., Zhang, Y. Y., Lu, Y., Zhu, D., Chen, Y. C., Liu, X. P., Wang, W. T., Wang, X. L., Liu, W. X., Li, B. G., and Piao, S. L.: High-resolution mapping of combustion processes and implications for CO2 emissions, Atmos. Chem. Phys., 13, 5189-5203, https://doi.org/5110.5194/acp-5113-5189-2013, 2013. Zheng, B., Tong, D., Li, M., Liu, F., Hong, C., Geng, G., Li, H., Li, X., Peng, L., Qi, J., Yan, L., Zhang, Y., Zhao, H., Zheng, Y., He, K., and Zhang, Q.: Trends in China's anthropogenic emissions since 2010 as the consequence of clean air actions, Atmos. Chem. Phys., 18, 14095-14111, https://doi.org/14010.15194/acp-14018-14095-12018, , 2018.

Please also note the supplement to this comment:
https://www.atmos-chem-phys-discuss.net/acp-2019-643/acp-2019-643-AC1-supplement.pdf

---

## Author Comment (AC2) · 30 Jul 2020

Responses to reviewer 2 Dear Editor and Reviewer, We thank you for your letter and for the reviewer's constructive comments concerning our manuscript. Those comments are all very valuable and helpful for revising and improving our paper. We have studied your comments carefully and have made corrections accordingly, which we hope to have addressed your concerns. Revised parts are marked in track change mode in the paper. The main corrections in the paper and the responses to the reviewer's comments are as follows.

Anonymous Referee #2 This is a potentially very interesting paper but the current version is poorly organized and inadequately explained. It needs serious reorganization to

[Figure]

tell a direct and clear story. Response: Thank you for your good suggestions, and we reorganized the results from national scale (total estimates in Section 4.1 and spatial distribution in Section 4.2), to provincial scale estimates and correlations, and finally to the finer grid level. Most of the graphics are quite adequate but need a bit more explanation. Figures4a, 4b, etc need a lot more explanation. Response: We added more descriptions and explanations in lines 297–300 for Figures 6a, 6b, 6d, 6g. The problems start early. This is a comparison of 9 datasets but sentence number 3 seems to accept values from one dataset (Le Quere et al.) – but this dataset does not appear to be part of the comparison. Response: Thank you for this question. Le Quere et al. dataset is named GCP/CDIAC in this comparison because their works in Global Carbon Project (GCP) used CDIAC data set for most years and used BP data to extrapolate the most recent two years. Sentence number 3 is a general background introduction due to its relatively large impact. Table 1 lists the properties of the datasets to be compared, but there are only 7 in the table. Response: Thank you for this question. We agree with you to complete Table 1 and added the other two datasets (GCP/CDIAC and NCCC) in Table 1. The original intention was to only include the gridded data that have been further analyzed for spatial characteristics in the latter part. Sentence number 2 of paragraph 2 jumps abruptly and without explanation from Chinese emissions estimates to global gridded emissions datasets. Response: Thank you, and we reorganized the introduction as total emissions estimate and spatial disaggregation. And we used transitional words to make the conjunction smoother. Line 58 introduces the CDIAC dataset, which also turns out to be not part of the comparison. Response: Thank you for this question. CDIAC is used by GCP and ODIAC, thus in total estimates they were identical for most of years, except for the recent two years that were extrapolated by BP data. And we added descriptions in lines 57-58, 136-137. In line 100 datasets from EIA, IEA, and BP are introduced, but also apparently not used in the comparison. There is no consistent story line on what is being compared and why, on the fact that comparison will be made at the national, provincial, and grid bases. Response: These three data sets do not include cement production emissions, and to make the data sets as comparable as

possible, we did not include them in the main text. Moreover, we showed them in the supplement (Figure S1) and pointed out this caveat. Table 1, text line 110, and Figure 2 all seem to say that CHRED exists only for 2007, but it appears elsewhere, for example Figure 4, with data for 2012? Response: Thank you for the careful review, revised. This is our overlook during data update. CHRED for year 2012 was just available in recent months through cooperation with data developer. The original comparison for CHRED was in 2007 and scaled to 2012 (Originally described in Figure 3 captions, and deleted after data update). Interesting but ad hoc statements appear throughout the text. Line 102 says that one of the purposes of the study was to identify "spatiotemporal differences" but there is no further mention of temporal differences. Response: Thank you for this question. We described the temporal differences in Section 4.1 (Figure 1) in lines around 168, 175, and 192. CARMA enters the discussion in line 109, without definition or citation. Response: Thank you. We added definition and citation in lines 304-310. EF enters the discussion in line 88 but if it is defined it is lost in a sea of acronyms. Response: Thank you, revised. Biomass burning appears in line 118 but there is no mention, until the closing discussion, on how it is used. Response: Thank you your question. Actually, only PKU included natural biomass burning from wild fire (Table S1, Emissions sectors), yet this only contributed a very small share close to 0, therefore it does not affect the estimates. I did not find enough discussion of Figure 1 to make it useful. Response: Thank you for this remind. We added in lines 146-149 and 334-336. Figure 1 is the summary of methodology for both total estimates and spatial disaggregation, i.e., activity data and EF determine the total emission estimates, and then affect the spatial distributions through disaggregation proxies of point, line and area sources. IF FCPSC was defined I missed it. Response: Sorry for the inconvenience. It first appeared in Table 1, and defined at the footnotes. We added it in the main text and also the acronym list. Line 139 says "both are 21%". But does not say % of what. This problem appears elsewhere in the text as well. Response: Sorry for the misleading. The range of the 9 estimates increased simultaneously from 0.7 to 2.1 Gt CO2, both of the ranges are 21% of the corresponding years' total emissions,

indicating the relative differences remained the same level. Also checked the others and revised in lines 111, 169 and 193. In line 299, "same" as what? Response: Thank you. Revised in line 393. The geolocation errors in China are relatively large, and only 45% of power plants were located in the same 0.1×0.1° grid in CARMA v2.0 as the real power plants locations that were identified by eyeballing in google maps (Fig. S1 in Wang t al, (2013)).

Text around lines 145 to 155 is so poorly organized that it is hard to follow. Response: Thank you. We reorganized and improved it by deleting trivial results and adding more explanations. Indeed, it is very challenging to explain all the differences among datasets, yet we provided the two main contributing factors: i.e., differences in EF for coal and systematic biases among national and provincial activity data. Page 10 is rambling and disconnected. On line 225 do I understand that total emissions from large point sources are approximately the same even though one data set has 2320 points and the other 945? How does this fit in with the 720, 1706, and 2320 in line 303? Response: Yes. Liu et al., (2015) reported that MEIC's power plant emissions are 2.5 Pg CO2 from 2320 power plants, while CARMA also estimated it 2.5 Pg CO2 from 945 plants (See below Fig. 13 from Liu et al., (2015)). As suggested in lines 308-309, The CARMA dataset does not provide accurate geolocations (latitude and longitude) (Byers et al., 2019) for the Chinese power plants and almost all inventories have corrected the original data and thus have different power plant numbers (Janssens-Maenhout et al., 2019;Liu et al., 2015;Liu et al., 2013;Wang et al., 2013). Moreover, EDGAR used CARMA3.0 while ODIAC and PKU used CARMA2.0, new version included more power plants.

I could list many additional problems of organization and flow of the text. There is much here that appears to be interesting. The paper needs a major re-organization and significant increase in explanations of what was done and why and what we learn from it. Response: Thank you. We re-organized the Introductions, Results and added more contents in discussions. We separated total emissions and spatial disaggregation through rewriting the 2nd and 3rd paragraphs. We arranged the introduction from a general background of China's fossil fuel $CO_2$ emissions, and then to the total estimates and spatial proxies, followed by the local inventories developed within China using more detailed provincial activity data and local optimized emission factors. Finally we pointed out the importance of this study: Why Chinese are possible more uncertain and why it is important.

References: Byers, L., Friedrich, J., Hennig, R., Kressig, A., Li, X., McCormick, C., and Malaguzzi, V. L.: A Global Database of Power Plants, in, World Resources Institute. Available online at www.wri.org/publication/global-database-power-plants., Washington, DC, 2019. Janssens-Maenhout, G., Crippa, M., Guizzardi, D., Muntean, M., Schaaf, E., Dentener, F., Bergamaschi, P., Pagliari, V., Olivier, J. G. J., Peters, J. A. H. W., van Aardenne, J. A., Monni, S., Doering, U., Petrescu, A. M. R., Solazzo, E., and Oreggioni, G. D.: EDGAR v4.3.2 Global Atlas of the three major greenhouse gas emissions for the period 1970–2012, Earth Syst. Sci. Data, 11, 959-1002, 10.5194/essd-11-959-2019, 2019. Liu, F., Zhang, Q., Tong, D., Zheng, B., Li, M., Huo, H., and He, K. B.: High-resolution inventory of technologies, activities, and emissions of coal-fired power plants in China from 1990 to 2010, Atmos. Chem. Phys., 15, 13299-13317, 2015. Liu, M., Wang, H., Oda, T., Zhao, Y., Yang, X., Zang, R., Zang, B., Bi, J., and Chen, J.: Refined estimate of China's $CO_2$ emissions in spatiotemporal distributions, Atmos. Chem. Phys., 13, 10873-10882, https://doi.org/10810.15194/acp-10813-10873-12013, 2013. Wang, R., Tao, S., Ciais, P., Shen, H. Z., Huang, Y., Chen, H., Shen, G. F., Wang, B., Li, W., Zhang, Y. Y., Lu, Y., Zhu, D., Chen, Y. C., Liu, X. P., Wang, W. T., Wang, X. L., Liu, W. X., Li, B. G., and Piao, S. L.: High-resolution mapping of combustion processes and implications for $CO_2$ emissions, Atmos. Chem. Phys., 13, 5189-5203, https://doi.org/5110.5194/acp-5113-5189-2013, 2013.

Please also note the supplement to this comment:
https://www.atmos-chem-phys-discuss.net/acp-2019-643/acp-2019-643-AC2-supplement.pdf

[Figure]

**Supplement:**

**Responses to reviewer 2**

Dear Editor and Reviewer,

We thank you for your letter and for the reviewer's constructive comments concerning our manuscript. Those comments are all very valuable and helpful for revising and improving our paper. We have studied your comments carefully and have made corrections accordingly, which we hope to have addressed your concerns. Revised parts are marked in track change mode in the paper. The main corrections in the paper and the responses to the reviewer's comments are as follows.

**Anonymous Referee #2**

This is a potentially very interesting paper but the current version is poorly organized and inadequately explained. It needs serious reorganization to tell a direct and clear story.

Response: Thank you for your good suggestions, and we reorganized the results from national scale (total estimates in Section 4.1 and spatial distribution in Section 4.2), to provincial scale estimates and correlations, and finally to the finer grid level.

Most of the graphics are quite adequate but need a bit more explanation. Figures4a, 4b, etc need a lot more explanation.

Response: We added more descriptions and explanations in lines 297–300 for Figures 6a, 6b, 6d, 6g.

The problems start early. This is a comparison of 9 datasets but sentence number 3 seems to accept values from one dataset (Le Quere et al.) – but this dataset does not appear to be part of the comparison.

Response: Thank you for this question. Le Quere et al. dataset is named GCP/CDIAC in this comparison because their works in Global Carbon Project (GCP) used CDIAC data set for most years and used BP data to extrapolate the most recent two years.

Sentence number 3 is a general background introduction due to its relatively large impact.

Table 1 lists the properties of the datasets to be compared, but there are only 7 in the table.

Response: Thank you for this question. We agree with you to complete Table 1 and added the other two datasets (GCP/CDIAC and NCCC) in Table 1. The original intention was to only include the gridded data that have been further analyzed for spatial characteristics in the latter part.

Sentence number 2 of paragraph 2 jumps abruptly and without explanation from Chinese emissions estimates to global gridded emissions datasets.

Response: Thank you, and we reorganized the introduction as total emissions estimate and spatial disaggregation. And we used transitional words to make the conjunction smoother.

Line 58 introduces the CDIAC dataset, which also turns out to be not part of the comparison.

Response: Thank you for this question. CDIAC is used by GCP and ODIAC, thus in total estimates they were identical for most of years, except for the recent two years that were extrapolated by BP data. And we added descriptions in lines 57-58, 136-137.

In line 100 datasets from EIA, IEA, and BP are introduced, but also apparently not used in the comparison. There is no consistent story line on what is being compared and why, on the fact that comparison will be made at the national, provincial, and grid bases.

Response: These three data sets do not include cement production emissions, and to make the data sets as comparable as possible, we did not include them in the main text. Moreover, we showed them in the supplement (Figure S1) and pointed out this caveat.

Table 1, text line 110, and Figure 2 all seem to say that CHRED exists only for 2007, but it appears elsewhere, for example Figure 4, with data for 2012?

Response: Thank you for the careful review, revised. This is our overlook during data update. CHRED for year 2012 was just available in recent months through

cooperation with data developer. The original comparison for CHRED was in 2007 and scaled to 2012 (Originally described in Figure 3 captions, and deleted after data update).

Interesting but ad hoc statements appear throughout the text. Line 102 says that one of the purposes of the study was to identify "spatiotemporal differences" but there is no further mention of temporal differences.

Response: Thank you for this question. We described the temporal differences in Section 4.1 (Figure 1) in lines around 168, 175, and 192.

CARMA enters the discussion in line 109, without definition or citation.

Response: Thank you. We added definition and citation in lines 304-310.

EF enters the discussion in line 88 but if it is defined it is lost in a sea of acronyms.

Response: Thank you, revised.

Biomass burning appears in line 118 but there is no mention, until the closing discussion, on how it is used.

Response: Thank you your question. Actually, only PKU included natural biomass burning from wild fire (Table S1, Emissions sectors), yet this only contributed a very small share close to 0, therefore it does not affect the estimates.

I did not find enough discussion of Figure 1 to make it useful.

Response: Thank you for this remind. We added in lines 146-149 and 334-336. Figure 1 is the summary of methodology for both total estimates and spatial disaggregation, i.e., activity data and EF determine the total emission estimates, and then affect the spatial distributions through disaggregation proxies of point, line and area sources.

IF FCPSC was defined I missed it.

Response: Sorry for the inconvenience. It first appeared in Table 1, and defined at the footnotes. We added it in the main text and also the acronym list.

Line 139 says "both are 21%". But does not say % of what. This problem appears elsewhere in the text as well.

Response: Sorry for the misleading. The range of the 9 estimates increased simultaneously from 0.7 to 2.1 Gt $CO_2$, both of the ranges are 21% of the corresponding years' total emissions, indicating the relative differences remained the

same level.

Also checked the others and revised in lines 111, 169 and 193.

In line 299, "same" as what?

Response: Thank you. Revised in line 393. The geolocation errors in China are relatively large, and only 45% of power plants were located in the same $0.1 \times 0.1°$ grid in CARMA v2.0 **as the real power plants locations that were identified by eyeballing in google maps (Fig. S1 in Wang t al, (2013))**.

[Figure]

Company: TRANSCANADA ENERGY LTD
Plant Name: GRANDVIEW
Location: Saint John, Canada
CARMA coordinate: 45.267°N, 66.067°W
Accurate coordinate: 45.279°N, 66.011°W

Company: CA ADMIN FOMENTO ELEC (CADAFE)
Plant Name: PLANTA CENTRO
Location: Moron, Venezuela
CARMA coordinate: 10.480°N, 68.200°W
Accurate coordinate: 10.495°N, 68.155°W

**Fig. S1**. Position offsets of randomly selected power stations recorded in the CARMA v2.0. The geographic positions of randomly selected 350 power stations (100 stations in China and 250 stations outside of China and U.S.A.) in the CARMA v2.0 list are checked against the presence of facility locations from visual inspection of Google imagery. The red circles are the true locations identified from Google Earth imagery, which are linked by blue lines to the CARMA v2.0 recorded locations. To do so, all stations in CARMA v2.0 were divided into 10 categories of equal sample sizes based on their annual fuel consumptions. For a stratified sampling, 50 stations (20 in China and 30 in other countries except the U.S.A.) were randomly selected from each category. The exact locations of the power stations were checked on Google Earth by searching the names of the stations and inspecting Google Earth images of power plants (chimneys and cooling towers). Roughly, 3 out of 4 stations selected were found in the Google Earth images, and 1 out of 4 stations could not be identified. As a result, 350 power stations with their locations (100 in China and 250 in other countries except the U.S.A.) were found after 476 stations were searched. The size of each circle is proportional to the emission from each power station. Two satellite images with typical views of power stations found on Google Earth are shown (the reported power stations by CARMA v2.0 are shown as red pentagrams).

Text around lines 145 to 155 is so poorly organized that it is hard to follow.

Response: Thank you. We reorganized and improved it by deleting trivial results and adding more explanations. Indeed, it is very challenging to explain all the differences among datasets, yet we provided the two main contributing factors: i.e., differences in EF for coal and systematic biases among national and provincial activity data.

Page 10 is rambling and disconnected. On line 225 do I understand that total emissions from large point sources are approximately the same even though one data set has 2320 points and the other 945? How does this fit in with the 720, 1706, and 2320 in line 303?

Response: Yes. Liu et al., (2015) reported that MEIC's power plant emissions are 2.5 Pg $CO_2$ from 2320 power plants, while CARMA also estimated it 2.5 Pg $CO_2$ from 945 plants (See below Fig. 13 from Liu et al., (2015)).

As suggested in lines 308-309, The CARMA dataset does not provide accurate geolocations (latitude and longitude) (Byers et al., 2019) for the Chinese power plants and almost all inventories have corrected the original data and thus have different power plant numbers (Janssens-Maenhout et al., 2019;Liu et al., 2015;Liu et al., 2013;Wang et al., 2013). Moreover, EDGAR used CARMA3.0 while ODIAC and PKU used CARMA2.0, new version included more power plants.

[Figure]

**Figure 13. (a)** Spatial distribution of $CO_2$ emissions in CPED in 2009. **(b)** Spatial distribution of $CO_2$ emissions in CARMA in 2009. **(c)** Comparisons of $CO_2$ emissions between CARMA and CPED by plant numbers in 2009. The plants are sorted according to ascending $CO_2$ emissions along the *y* axis. The red and blue lines denote the plant number cumulative ratio for CARMA and CPED, respectively. **(d)** Comparisons of the spatial distribution of $CO_2$ emissions in southwest China between CARMA and CPED in 2009.

I could list many additional problems of organization and flow of the text. There is much here that appears to be interesting. The paper needs a major re-organization and significant increase in explanations of what was done and why and what we learn from it.

Response: Thank you. We re-organized the Introductions, Results and added more contents in discussions. We separated total emissions and spatial disaggregation through rewriting the 2$^{nd}$ and 3$^{rd}$ paragraphs. We arranged the introduction from a general background of China's fossil fuel $CO_2$ emissions, and then to the total estimates and spatial proxies, followed by the local inventories developed within China using more detailed provincial activity data and local optimized emission factors. Finally we pointed out the importance of this study: Why Chinese are possible more uncertain and why it is important.

References:

Byers, L., Friedrich, J., Hennig, R., Kressig, A., Li, X., McCormick, C., and Malaguzzi, V. L.: A Global Database of Power Plants, in, World Resources Institute. Available online at www.wri.org/publication/global-database-power-plants., Washington, DC, 2019.

Janssens-Maenhout, G., Crippa, M., Guizzardi, D., Muntean, M., Schaaf, E., Dentener, F., Bergamaschi, P., Pagliari, V., Olivier, J. G. J., Peters, J. A. H. W., van Aardenne, J. A., Monni, S., Doering, U., Petrescu, A. M. R., Solazzo, E., and Oreggioni, G. D.: EDGAR v4.3.2 Global Atlas of the three major greenhouse gas emissions for the period 1970–2012, Earth Syst. Sci. Data, 11, 959-1002, 10.5194/essd-11-959-2019, 2019.

Liu, F., Zhang, Q., Tong, D., Zheng, B., Li, M., Huo, H., and He, K. B.: High-resolution inventory of technologies, activities, and emissions of coal-fired power plants in China from 1990 to 2010, Atmos. Chem. Phys., 15, 13299-13317, 2015.

Liu, M., Wang, H., Oda, T., Zhao, Y., Yang, X., Zang, R., Zang, B., Bi, J., and Chen, J.: Refined estimate of China's $CO_2$ emissions in spatiotemporal distributions, Atmos. Chem. Phys., 13, 10873-10882, https://doi.org/10810.15194/acp-10813-10873-12013, 2013.

Wang, R., Tao, S., Ciais, P., Shen, H. Z., Huang, Y., Chen, H., Shen, G. F., Wang, B., Li, W., Zhang, Y. Y., Lu, Y., Zhu, D., Chen, Y. C., Liu, X. P., Wang, W. T., Wang, X. L., Liu, W. X., Li, B. G., and Piao, S. L.: High-resolution mapping of combustion processes and implications for $CO_2$ emissions, Atmos. Chem. Phys., 13, 5189-5203, https://doi.org/5110.5194/acp-5113-5189-2013, 2013.

---

## Author Response (AR2)

Responses to Editor

Dear Editor,

We thank you for your constructive comments and all the thoughtful and hard work during the handling of this paper. Those comments are very valuable and helpful for revising and improving this paper. We have made corrections accordingly, which we hope to have addressed your concerns. Revised parts are marked in track change mode in the paper. The main corrections are as follows.

Comments to the Author:
There are grammar errors such as Line 145 and other lines. The authors need to check all grammar instead of this current form.

Response: Thank you for this suggestion. We have carefully revised the grammar errors and sent the MS to be polished by a native speaker. And we corrected all the grammar errors as can be seen throughout the MS.

Line 85, EDGAR

Response: Thank you, revised accordingly.

Line 100, the first author should be Zhang, et al. (2007). Authors should not rely on EndNote that usually brought errors for the references and checks the accuracy for all references.

Response: Thank you for this careful check and good suggestion. We have carefully double-checked and corrected all the wrong formats of references, such as the corrections in lines 103, and 334 for Zhang et al., (2007).

Line 122, authors deleted the EIA, IEA, and BP. Are they one of 9 global and regional emission datasets at Line 34? Will the number of inventories not be 9 emission datasets in the abstract if they are not compared with others? it may be a bit confusing. The authors need to explain it for the later sections that mentioned them again.

Response: Thank you for this good suggestion. We added descriptions in lines 132-133 to clearly define the 9 inventories. EIA, IEA, and BP do not belong to the 9

inventories. The 9 inventories used in the main text include: 1) 6 gridded datasets, namely ODIAC, EDGAR, PKU, CHRED, MEIC, NJU, and 3 statistical data, namely GCP/CDIAC, CEADs, and NCCC.